# All-atom Diffusion Transformers:
# Unified generative modelling of molecules and materials

**Chaitanya K. Joshi** [1 2]   **Xiang Fu** [1]   **Yi-Lun Liao** [1 3]   **Vahe Gharakhanyan** [1]   **Benjamin Kurt Miller** [1]
**Anuroop Sriram** [* 1]   **Zachary W. Ulissi** [* 1]

## Abstract

Diffusion models are the standard toolkit for generative modelling of 3D atomic systems. However, for different types of atomic systems – such as molecules and materials – the generative processes are usually highly specific to the target system despite the underlying physics being the same. We introduce the All-atom Diffusion Transformer (ADiT), a unified latent diffusion framework for jointly generating both periodic materials and non-periodic molecular systems using the same model: (1) An autoencoder maps a unified, all-atom representations of molecules and materials to a shared latent embedding space; and (2) A diffusion model is trained to generate new latent embeddings that the autoencoder can decode to sample new molecules or materials. Experiments on MP20, QM9 and GEOM-DRUGS datasets demonstrate that jointly trained ADiT generates realistic and valid molecules as well as materials, obtaining state-of-the-art results on par with molecule and crystal-specific models. ADiT uses standard Transformers with minimal inductive biases for both the autoencoder and diffusion model, resulting in significant speedups during training and inference compared to equivariant diffusion models. Scaling ADiT up to half a billion parameters predictably improves performance, representing a step towards broadly generalizable foundation models for generative chemistry. Open source code: https://github.com/facebookresearch/all-atom-diffusion-transformer

---
[*]Equal contribution [1]Fundamental AI Research (FAIR) at Meta [2]University of Cambridge [3]MIT. Correspondence to: Chaitanya K. Joshi <chaitanya.joshi@cl.cam.ac.uk>.

*Proceedings of the 42nd International Conference on Machine Learning*, Vancouver, Canada. PMLR 267, 2025. Copyright 2025 by the author(s).

## 1. Introduction

Generative modelling of the 3D structure of atomic systems has the potential to revolutionize inverse design of new molecules and materials. The current state-of-the-art uses diffusion or flow matching models for tasks such as structure prediction (Abramson et al., 2024; Corso et al., 2023; Jiao et al., 2023) and conditional generation (Watson et al., 2023; Ingraham et al., 2023; Zeni et al., 2025) for biomolecules and materials, as well as for structure-based drug design (Schneuing et al., 2024).

All atomic systems share the same underlying physical principles that determine their 3D structure and interactions. However, we currently do not have a unified formulation of diffusion models across different types of atomic systems such as small molecules, biomolecules, crystals, and their combinations. Most diffusion models are highly specific to each type of system, and involve multi-modal generative processes on complex product manifolds of categorical and continuous data types. For example, de novo generation of small molecules is modelled as two independent diffusion processes for the atom types (categorical) and 3D coordinates (continuous) of a set of atoms (Hoogeboom et al., 2022). The denoiser model learns how atom types and 3D coordinates jointly evolve in order to sample new molecules but passes through unrealistic intermediate states during the denoising trajectory. Diffusion models for biomolecules treat groups of atoms as rigid bodies and add a third manifold (rotations) into the joint diffusion process (Yim et al., 2023b; Campbell et al., 2024). For crystals and materials, the diffusion process needs to additionally handle periodicity and operates on a joint manifold of atom types, fractional coordinates, lattice lengths, and lattice angles that together define the repeating unit cell (Xie et al., 2022; Miller et al., 2024).

In this paper, we pose the following question: *How can we build unified diffusion models that can generate both periodic materials and non-periodic molecular systems?*

Our solution, the **All-atom Diffusion Transformer (ADiT)**, illustrated in Figure 1, is a latent diffusion model based on two key ideas:

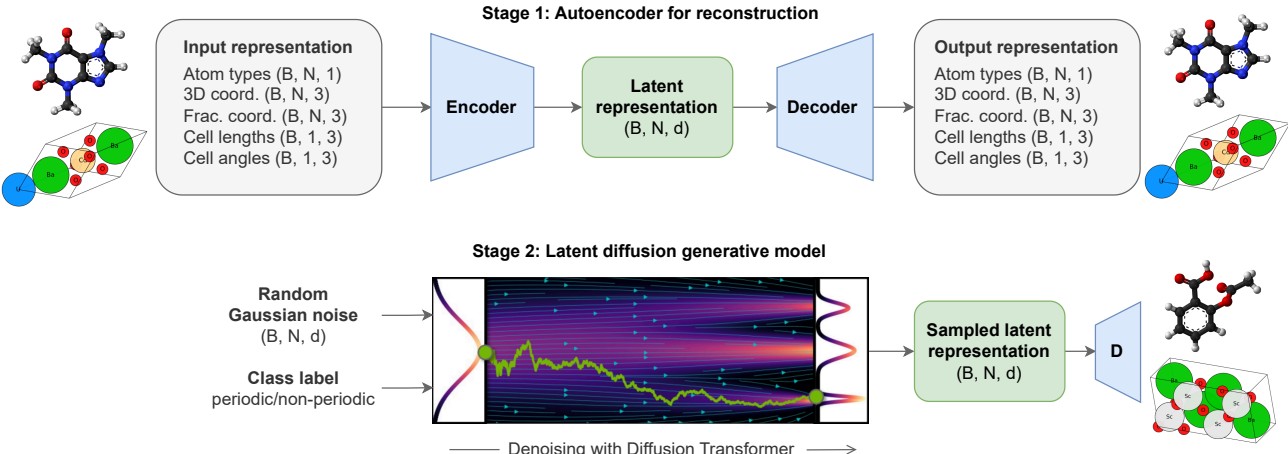

Figure 1: **Unified generative modelling of molecules and materials with All-atom Diffusion Transformers.** ADiT performs generative modelling of chemical systems in two stages: (1) A Variational Autoencoder (VAE) learns a shared latent space by reconstructing all-atom representations of both molecules (non-periodic) and crystals (periodic); and (2) A Diffusion Transformer (DiT) samples new latents from the shared distribution using classifier-free guidance, which are decoded to valid molecules or crystals using the VAE. Our unified latent diffusion framework enables transfer learning and avoids the complexity of multiple diffusion processes on categorical-continuous product manifolds used by equivariant diffusion models.

1. *All-atom unified latent representations:* We treat both periodic and non-periodic atomic systems as sets of atoms in 3D space and develop a unified representation with categorical and continuous attributes per atom. A Variational Autoencoder (VAE) (Kingma & Welling, 2014) embeds molecules and crystals into a shared latent space by training for all-atom reconstruction.

2. *Latent diffusion using Transformers:* We perform generative modelling in the latent space of the VAE encoder using a Diffusion Transformer (DiT) (Rombach et al., 2022; Peebles & Xie, 2023). During inference, classifier-free guidance (Ho & Salimans, 2022) enables sampling new latents that can be reconstructed to valid molecules or crystals using the VAE decoder.

ADiTs can be trained jointly on both periodic and non-periodic atomic systems, demonstrating broad generalizability. Training a single unified model on the QM9 molecular and MP20 materials datasets leads to state-of-the-art performance in both domains, exceeding specialized equivariant diffusion models on physics-based validations. DFT calculations reveal that ADiTs generate stable, unique, and novel crystals at a 5-6% S.U.N. rate, a 25% improvement upon the 4-5% rates of previous methods. Joint training yields higher validity rates than QM9-only or MP20-only ADiT variants, demonstrating successful transfer learning between periodic and non-periodic systems. ADiTs also match or exceed state-of-the-art equivariant models on the GEOM-DRUGS dataset of molecules with hundreds of atoms.

ADiTs are highly scalable, achieving significant speedups in both training and inference compared to equivariant diffusion models. By using standard Transformers with minimal inductive biases for both the autoencoder and diffusion model, ADiTs can generate 10,000 samples in under 20 minutes on a single V100 GPU – an order of magnitude faster than baselines which take up to 2.5 hours on the same hardware. The practical efficiency of the DiT denoiser compared to equivariant networks allows us to scale ADiT to half a billion parameters while keeping data scale fixed. Our scaling law analysis demonstrates that generative modelling performance improves predictably with model size, suggesting further gains are possible through continued scaling.

All together, our work is the first to develop unified generative models for both periodic and non-periodic atomic systems, with state-of-the-art performance on both molecules and crystals. ADiTs represent a step towards broadly generalizable foundation models for generative chemistry.

## 2. All-atom Diffusion Transformers

**Overview.** We use latent diffusion (Rombach et al., 2022) to unify generative modelling across periodic and non-periodic atomic systems. Our approach consists of two stages: (1) A Variational Autoencoder (VAE) (Kingma & Welling, 2014) learns a shared latent space by jointly reconstructing all-atom representations of both molecules and materials; and (2) A Diffusion Transformer (Peebles & Xie, 2023) generates new samples from this latent space

which can be decoded into valid molecules or crystals using classifier-free guidance (Ho & Salimans, 2022). Latent diffusion shifts the complexity of handling categorical and continuous attributes into the autoencoder, enabling a simplified and scalable generative process in latent space. We discuss how our contributions are contextualized w.r.t. related work in Appendix A.

## 2.1. Stage 1: Autoencoder for reconstruction

**Unified representation of 3D atomic systems.** All periodic and non-periodic atomic systems can be represented in a unified format as sets of atoms in 3D space (Duval et al., 2023). The key difference is that crystals require an additional periodic unit cell, while molecules have unbounded coordinates. A crystal or molecule with $N$ atoms is represented as a multi-modal object:

$$
\begin{aligned}
\text{Atom types } \boldsymbol{A} = \{a_i\}_{i=1}^N &\in \mathbb{Z}^{1\times N}, \\
\text{3D coordinates } \boldsymbol{X} = \{x_i\}_{i=1}^N &\in \mathbb{R}^{3\times N}, \\
\text{Fractional coordinates } \boldsymbol{F} = \{f_i\}_{i=1}^N &\in [0,1)^{3\times N}, \\
\text{Unit cell/lattice } \boldsymbol{L} = \{l_1, l_2, l_3\} &\in \mathbb{R}^{3\times 3}.
\end{aligned}
$$

The 3D coordinates $\boldsymbol{X}$ are in nanometers, and the fractional coordinates $\boldsymbol{F}$ are in the range $[0,1)$. The lattice matrix $\boldsymbol{L}$ represents a parallelepiped defining the shape of the repeating unit cell, and fractional coordinates are computed as the inverse of the unit cell matrix multiplied by the 3D coordinates: $\boldsymbol{F} = \boldsymbol{L}^{-1}\boldsymbol{X}$. We use Niggli reduction to uniquely determine the unit cell parameters for crystals (Grosse-Kunstleve et al., 2004). For non-periodic molecules, we set the unit cell parameters and fractional coordinates to null values $\phi$.

**VAE architecture.** We use a Variational Autoencoder (VAE) to learn a shared latent representation of molecules and materials using a reconstruction objective. Given an input 3D atomic system $(\boldsymbol{A}, \boldsymbol{X}, \boldsymbol{F}, \boldsymbol{L})$, an encoder $\mathcal{E}$ maps each atom's attributes to a latent representation $\boldsymbol{Z}$:

$$\boldsymbol{Z} = \mathcal{E}(\boldsymbol{A}, \boldsymbol{X}, \boldsymbol{F}), \tag{1}$$

where $\boldsymbol{Z} = \{z_i\}_{i=1}^N \in \mathbb{R}^{d\times N}$ encodes information about the categorical atom type and continuous coordinates (unit cell parameters are encoded implicitly in the fractional coordinates). The decoder $\mathcal{D}$ reconstructs the input atomic system from the latent embedding:

$$\boldsymbol{A}', \boldsymbol{X}', \boldsymbol{F}', \boldsymbol{L}' = \mathcal{D}(\boldsymbol{Z}). \tag{2}$$

We describe the pseudocode for VAE encoder and decoder operations in Algorithms 1 and 2, respectively. For the architecture of the encoder $\mathcal{E}$ and decoder $\mathcal{D}$, we used the standard Transformer (Vaswani et al. (2017), `torch.nn.TransformerEncoder`) and learn symmetries

via data augmentation. In Appendix D, we also ablated roto-translation equivariant VAEs based on Equiformer-V2 (Liao et al., 2024).

---

**Algorithm 1:** Pseudocode for VAE encoder $\mathcal{E}$

---

**Input:** 3D atomic system $(\{a_i\}, \{x_i\}, \{f_i\}, \{l_1, l_2, l_3\})$
**Output:** Latent reprenstations $\{z_i\}$
    # Project inputs to $d_{\text{model}}$
1.    $h_i = \text{Embedding}(a_i)$          $h_i \in \mathbb{R}^{d_{\text{model}}}$
2.    $h_i = h_i + \text{Linear}(\text{Swish}(\text{Linear}(x_i)))$
3.    $h_i = h_i + \text{Linear}(\text{Swish}(\text{Linear}(f_i)))$
    # Apply encoder network
4.    $\{h_i\} = \text{TransformerEncoder}(\{h_i\})$
    # Down-project to mean $\mu_{\boldsymbol{Z}}$ and std $\sigma_{\boldsymbol{Z}}$
5.    $\mu_{z_i} = \text{Linear}(h_i)$          $\mu_{z_i} \in \mathbb{R}^d$
6.    $\log \sigma_{z_i} = \text{Linear}(h_i)$      $\sigma_{z_i} \in \mathbb{R}^d$
    # Sample latents $\boldsymbol{Z}$
7.    $z_i = \mu_{z_i} + \sigma_{z_i} \odot \epsilon, \quad \epsilon \sim \mathcal{N}(0,1)^d$    $z_i \in \mathbb{R}^d$

---

**Algorithm 2:** Pseudocode for VAE decoder $\mathcal{D}$

---

**Input:** Latent reprenstations $\{z_i\}$
**Output:** 3D atomic system $(\{a_i'\}, \{x_i'\}, \{f_i'\}, \{l_1', l_2', l_3'\})$
    # Up-project latents to $d_{\text{model}}$
1.    $h_i = \text{Linear}(z_i)$          $h_i \in \mathbb{R}^{d_{\text{model}}}$
    # Apply decoder network
2.    $\{h_i\} = \text{TransformerEncoder}(\{h_i\})$
    # Predict outputs
3.    $a_i' = \text{argmax}(\text{Linear}(h_i))$      $a_i' \in \mathbb{Z}$
4.    $x_i' = \text{Linear}(h_i)$          $x_i' \in \mathbb{R}^3$
5.    $f_i' = \text{Linear}(h_i)$          $f_i' \in \mathbb{R}^3$
6.    $\{l_1', l_2', l_3'\} = \text{Linear}\left(\frac{1}{N}\sum_{i=1}^N h_i\right)$    $l' \in \mathbb{R}^3$

---

**Reconstruction loss.** We compute the loss for the predicted atom types $\boldsymbol{A}'$ via cross-entropy:

$$\mathcal{L}_{\boldsymbol{A}} = \frac{1}{N}\sum_{i=1}^N \text{CrossEnt}(a_i, a_i'). \tag{3}$$

For the predicted 3D coordinates $\boldsymbol{X}'$, we use the mean squared error (MSE) reconstruction loss after zero-centering both sets of coordinates:

$$\tilde{x}_i = x_i - \frac{1}{N}\sum_{i=1}^N x_i, \quad \tilde{x}_i' = x_i' - \frac{1}{N}\sum_{i=1}^N x_i',$$

$$\mathcal{L}_{\boldsymbol{X}} = \frac{1}{3N}\sum_{i=1}^N \|\tilde{x}_i - \tilde{x}_i'\|^2. \tag{4}$$

We compute the reconstruction loss for the predicted fractional coordinates $\boldsymbol{F}'$ using MSE as well:

$$\mathcal{L}_{\boldsymbol{F}} = \frac{1}{3N}\sum_{i=1}^N \|f_i - f_i'\|^2. \tag{5}$$

For the predicted lattice vectors $\boldsymbol{L}'$, we first convert to rotation-invariant lattice parameters: three side lengths of

the unit cell $\boldsymbol{L}_l = \{a, b, c\} \in \mathbb{R}^{1\times3}$, and three internal angles between them $\boldsymbol{L}_a = \{\alpha, \beta, \gamma\} \in [60°, 120°]^{1\times3}$, as described in Miller et al. (2024). We then compute the MSE reconstruction loss between the predicted and ground truth lattice parameters:

$$\mathcal{L}_{\boldsymbol{L}_l} = \frac{1}{3} \left( (a - a')^2 + (b - b')^2 + (c - c')^2 \right), \quad (6)$$

$$\mathcal{L}_{\boldsymbol{L}_a} = \frac{1}{3} \left( (\alpha - \alpha')^2 + (\beta - \beta')^2 + (\gamma - \gamma')^2 \right). \quad (7)$$

Note that in $\mathcal{L}_{\boldsymbol{L}_l}$, we normalize the predicted and groundtruth lengths by the cube root of the number of atoms to account for the scaling of the unit cell with the number of atoms, following Xie et al. (2022). All angles are converted from degree to radians for numerical stability.

The autoencoder is trained with a weighted reconstruction loss to balance the relative magnitudes of the various losses:

$$\mathcal{L}_{\text{rec}} = \lambda_{\boldsymbol{A}}\mathcal{L}_{\boldsymbol{A}} + \lambda_{\boldsymbol{X}}\mathcal{L}_{\boldsymbol{X}} + \lambda_{\boldsymbol{F}}\mathcal{L}_{\boldsymbol{F}} + \lambda_{\boldsymbol{L}_l}\mathcal{L}_{\boldsymbol{L}_l} + \lambda_{\boldsymbol{L}_a}\mathcal{L}_{\boldsymbol{L}_a}, \quad (8)$$

Depending on whether a training sample is periodic or non-periodic, we use different reconstruction loss weights:

| | $\lambda_{\boldsymbol{A}}$ | $\lambda_{\boldsymbol{X}}$ | $\lambda_{\boldsymbol{F}}$ | $\lambda_{\boldsymbol{L}_l}$ | $\lambda_{\boldsymbol{L}_a}$ |
|---|---|---|---|---|---|
| Periodic | 1.0 | 0.0 | 10.0 | 1.0 | 10.0 |
| Non-periodic | 1.0 | 10.0 | 0.0 | 0.0 | 0.0 |

Thus, the overall loss for periodic crystals trains the model to reconstruct the atom types, fractional coordinates and lattice parameters while ignoring the predicted 3D coordinates. Similarly, the overall loss for non-periodic molecules trains the model to reconstruct the atom types and 3D coordinates while ignoring the predicted fractional coordinates and lattice parameters.

**Regularization.** We use three regularization techniques to learn robust, informative latent representations: (1) A bottleneck architecture with latent dimension $d$ significantly smaller than the encoder/decoder hidden dimension $d_{\text{model}}$ (e.g., $d = 8$ vs $d_{\text{model}} = 512$). (2) A per-channel KL divergence penalty $\lambda_{\text{KL}} \cdot D_{\text{KL}}( \mathcal{N}(\boldsymbol{Z}; \mu_{\boldsymbol{Z}}, \sigma_{\boldsymbol{Z}}) \mid\mid \mathcal{N}(0, 1)^d )$ added to equation 8, following Rombach et al. (2022). (3) Denoising training with 10% of atoms having their types masked and coordinates perturbed by $\mathcal{N}(0, 0.1)$ Gaussian noise. For non-equivariant encoders/decoders, we also randomly rotate and translate each sample during training to learn symmetries via data augmentation.

**Decoding latents to atomic systems.** During inference or sampling from the DiT, the desired output type (periodic/non-periodic) determines how we process the decoder outputs. The VAE decoder $\mathcal{D}$ generates four attributes for each system: (1) atom types, (2) 3D coordinates, (3) fractional coordinates, and (4) lattice parameters. For non-periodic molecules, we only utilize the atom types and 3D

coordinates, constructing the molecule via RDKit. For periodic crystals, we combine the atom types, fractional coordinates, and lattice parameters to build the crystal structure using PyMatGen. This split decoding strategy allows a single unified model to share information between both domains while still respecting their distinct geometric constraints, enabling effective transfer learning between periodic and non-periodic systems.

### 2.2. Stage 2: Latent diffusion generative model

**Diffusion formulation.** We use Gaussian diffusion or flow matching as our generative framework, which iteratively denoises latent samples from a base distribution into samples from a target distribution (Sohl-Dickstein et al., 2015; Song & Ermon, 2019; Ho et al., 2020; Lipman et al., 2023). Our formulation uses linear interpolation between a standard normal base distribution and the target distribution of VAE encoder latent representations of 3D atomic systems (we describe it in terms of flow matching, though both formulations are equivalent; see Gao et al. (2024)). Thus, the diffusion model is trained after training the first stage VAE.

Our model learns to generate a set of $N$ latent representations $\boldsymbol{Z} = \{z_i\}_{i=1}^N$, where each latent $z \in \mathbb{R}^d$ encodes information about one atom's type, coordinates and unit cell, which can be decoded to a valid 3D atomic system using the VAE decoder $\mathcal{D}$. During training, given an input 3D atomic system $(\boldsymbol{A}, \boldsymbol{X}, \boldsymbol{F}, \boldsymbol{L})$, we first encode it to a latent representation $\boldsymbol{Z}$ using the VAE encoder $\mathcal{E}$. We denote $\boldsymbol{Z}$ as $\boldsymbol{Z}^{(1)}$, a 'clean' training sample at time $t = 1$. We then sample a random initial latent $\boldsymbol{Z}^{(0)}$ at time $t = 0$ from a $d$-dimensional standard normal distribution $\mathcal{N}(0, 1)^d$, and perform zero-centering by subtracting the per-channel mean of $\boldsymbol{Z}^{(0)}$. We then use linear interpolation to construct a 'noisy' interpolated sample $\boldsymbol{Z}^{(t)}$ at a randomly sampled time step $t \sim \mathcal{U}(0, 1)$:[1]

$$\boldsymbol{Z}^{(t)} = (1 - t)\, \boldsymbol{Z}^{(0)} + t\, \boldsymbol{Z}^{(1)}. \quad (9)$$

Thus, we can define a groundtruth conditional vector field $u_t(\boldsymbol{Z}^{(t)} | \boldsymbol{Z}^{(1)})$ along the path from the noisy latents $\boldsymbol{Z}^{(t)}$ at time step $t$ to the clean latents $\boldsymbol{Z}^{(1)}$ as:

$$u_t(\boldsymbol{Z}^{(t)} | \boldsymbol{Z}^{(1)}) = \frac{\boldsymbol{Z}^{(1)} - \boldsymbol{Z}^{(t)}}{1 - t}. \quad (10)$$

Samples from the base distribution can be transformed to samples from the target distribution by integrating the vector field $u_t(\boldsymbol{Z}^{(t)} | \boldsymbol{Z}^{(1)})$ over time $t$.

The goal of conditional flow matching is to train a denoiser network $\mathcal{F}$ to match this conditional vector field $u_t$. To do so, the denoiser takes as input the intermediate noisy latents $\boldsymbol{Z}^{(t)}$ at time step $t$ and an additional class label $c$ (described

---

[1]In practice, we set a minimum value for time step $t_{\text{min}} = 0.01$.

subsequently) to predict the final clean latents $\boldsymbol{Z}'^{(1)}$:

$$\boldsymbol{Z}'^{(1)} = \mathcal{F}(\boldsymbol{Z}^{(t)}, t, c) \, . \tag{11}$$

The denoiser is trained by minimizing an MSE loss between the resulting predicted conditional vector field and the groundtruth conditional vector field:

$$\mathcal{L}_{\text{fm}} = \frac{1}{N} \sum_{i=1}^{N} \left\| \frac{z_i^{(1)} - z_i^{(t)}}{1-t} - \frac{z_i'^{(1)} - z_i^{(t)}}{1-t} \right\|^2 , \tag{12}$$

$$= \frac{1}{(1-t)^2} \frac{1}{N} \sum_{i=1}^{N} \| z_i^{(1)} - z_i'^{(1)} \|^2 \, .$$

In practice, we follow Yim et al. (2023a) and clip the value of $t$ at 0.9 to prevent numerical instability.

**Denoiser architecture.** As the denoiser network $\mathcal{F}$, we use a class-conditional Diffusion Transformer (DiT) (Peebles & Xie, 2023). The DiT largely follows a standard Transformer architecture with the conditioning information incorporated via adaptive layer norm with zero-initialization, which replaces all layer norm operations. For class conditioning, we use a binary embedding to denote whether the system being generated is periodic (crystal) or non-periodic (molecule). This conditioning allows the model to learn domain-specific features while sharing most parameters. During training, we apply class label dropout with 10% probability to enable classifier-free guidance during inference. We also incorporate self-conditioning (Yim et al., 2023b) where the denoiser's prediction from the previous timestep is concatenated to the current input with 50% dropout probability during training. While we currently only condition on the periodic/non-periodic class label, the DiT architecture can incorporate additional conditioning signals like target properties or geometric constraints to enable controlled generation. This represents a promising direction for future work in inverse design applications.

**Data augmentation.** The DiT denoiser is trained with data augmentation to learn roto-translational and periodic symmetries in the VAE's latent space. During training, each input system coordinates are randomly rotated and translated, and then converted to latents via the frozen VAE encoder $\mathcal{E}$ before being input to the DiT.

**Sampling with classifier-free guidance.** To generate new atomic systems from the trained diffusion model, we use classifier-free guidance (Ho & Salimans, 2022) to steer the sampling process. At each denoising step, we compute both a conditional prediction based on the periodic/non-periodic class label $c$ and an unconditional prediction with null class label $\phi$. The final prediction is a weighted combination of these using guidance scale $\gamma$, allowing control over how strongly the generation follows the class conditioning. The full sampling procedure is outlined in Algorithm 3. Starting

from Gaussian noise $\boldsymbol{Z}^{(0)}$, we iteratively denoise using the DiT model $\mathcal{F}$ for $T$ steps. At each step, we perform Euler integration of the vector field to gradually transform the noisy latents towards the target distribution. While we currently use simple Euler integration for efficiency, adaptive ODE solvers could potentially improve performance (Ma et al., 2024). Finally, we decode the denoised latents $\boldsymbol{Z}^{(1)}$ to a valid 3D atomic system using the VAE decoder $\mathcal{D}$.

---

**Algorithm 3:** Pseudocode for DiT sampling

---

**Input:** Class label $c$, num. integration steps $T$, cfg. scale $\gamma$
**Output:** Generated sample $(\boldsymbol{A}, \boldsymbol{X}, \boldsymbol{F}, \boldsymbol{L})$

    # Sample initial noisy latents $\boldsymbol{Z}^{(0)}$ at $t = 0$
1.  $\boldsymbol{Z}^{(0)} = \{z_i^{(0)} \sim \mathcal{N}(0,1)^d\}$
2.  $\Delta t = 1/T$     # Step size
    # Denoising loop
3.  for $t$ in linspace(0.0, 1.0, $T$):
4.    $\boldsymbol{Z}'_{\text{cond}} = \mathcal{F}(\boldsymbol{Z}^{(t)}, t, c)$   # Conditional prediction
5.    $\boldsymbol{Z}'_{\text{uncond}} = \mathcal{F}(\boldsymbol{Z}^{(t)}, t, \phi)$   # Unconditional prediction
    # Conditioning via classifier-free guidance
6.    $\boldsymbol{Z}' = (1 - \gamma) \cdot \boldsymbol{Z}'_{\text{uncond}} + \gamma \cdot \boldsymbol{Z}'_{\text{cond}}$
    # Euler integration step
7.    $\boldsymbol{Z}^{(t+\Delta t)} = \boldsymbol{Z}^{(t)} + \Delta t \cdot \frac{\boldsymbol{Z}' - \boldsymbol{Z}^{(t)}}{1-t}$
    # Decode latents to 3D atomic system (Algorithm 2)
8.  $\boldsymbol{A}, \boldsymbol{X}, \boldsymbol{F}, \boldsymbol{L} = \mathcal{D}(\boldsymbol{Z}^{(1)})$

---

## 3. Experimental Setup

**Datasets.** For our main experiments, we train models on periodic crystals from MP20 and non-periodic molecules from QM9, representing two distinct domains of atomic systems. MP20 (Xie et al., 2022) contains 45,231 metastable crystal structures from the Materials Project (Jain et al., 2013), each with up to 20 atoms in its unit cell and spanning 89 different element types. QM9 (Wu et al., 2018) consists of 130,000 stable small organic molecules containing up to nine heavy atoms (C, N, O, F) along with hydrogens. We split the data following prior work (Xie et al., 2022; Hoogeboom et al., 2022) to ensure fair comparisons. We also include results on the GEOM-DRUGS dataset of 430,000 large organic molecules up to 180 atoms (Axelrod & Gomez-Bombarelli, 2022), as well as the QMOF dataset of 14,000 metal-organic framework structures (Rosen et al., 2021).

**Training and hyperparameters.** We sequentially train the first-stage VAE and then the second-stage DiT using AdamW optimizer with a constant learning rate $1e - 4$, no weight decay, and batch size of 256. We use exponential moving average (EMA) of DiT weights over training with a decay of 0.9999. Both models are trained to convergence for at most 5000 epochs up to 3 days on 8 V100 GPUs.

For the first-stage VAE, we use a standard Transformer as both encoder $\mathcal{E}$ and decoder $\mathcal{D}$ with hidden dimension

Table 1: **Crystal generation results on MP20.** We report validity, stability, uniqueness, and novelty rates for 10,000 sampled crystals. ADiT shows improved performance over diffusion baselines across all metrics. We see significant gains for compositional validity due to a single diffusion process in the latent space, as opposed to joint continuous and categorical diffusion for baselines. Joint training with both molecular and crystal data improves crystal generation performance over MP20-only models. (Stable: DFT $E^{hull} < 0.0$, metastable: DFT $E^{hull} < 0.1$, * denotes results from MatterGen-MP for 1024 sampled crystals, $\dagger$ denotes results we replicated using the same DFT setup as ADiT.)

| | Model | Validity Rate (%) ↑ | | | Metastable rate (%) ↑ | Stable rate (%) ↑ | M.S.U.N. rate (%) ↑ | S.U.N. rate (%) ↑ |
| | | Structure | Composition | Overall | | | | |
|---|---|---|---|---|---|---|---|---|
| MP20-only | CDVAE | 100.00 | 86.70 | - | - | 1.6 | - | - |
| | DiffCSP | 100.00 | 83.25 | - | - | 5.0 | - | 3.3 |
| | UniMat | 97.2 | 89.4 | - | - | - | - | - |
| | FlowMM | 96.85 | 83.19 | 80.30 | $30.6^\dagger$ | $4.6^\dagger$ | $22.5^\dagger$ | $2.8^\dagger$ |
| | FlowLLM | 99.94 | 90.84 | 90.81 | $66.9^\dagger$ | $13.9^\dagger$ | $26.3^\dagger$ | $4.7^\dagger$ |
| | MatterGen-MP | - | - | - | 78* | 13* | 21* | - |
| | MP20-only ADiT | 99.58 | 90.46 | 90.13 | 81.6 | 14.1 | 25.91 | 4.7 |
| | Jointly trained ADiT | 99.74 | 92.14 | 91.92 | 81.0 | 15.4 | 28.2 | 5.3 |

$d_{model} = 512$, 8 attention heads, and 8 layers (51M parameters total). The latent dimension is set to $d = 8$ with KL regularization weight $\lambda_{KL} = 1e-5$ and 10% denoising perturbation during training. For the second-stage DiT denoiser, we report results primarily using DiT-B configurations: hidden dimension $d_{model} = 768$, 12 attention heads, 12 layers, and 130M parameters total. We also evaluate smaller DiT-S (32M parameters) and larger DiT-L (450M) variants.

Two key inference-time hyperparameters are the number of ODE integration steps $T$ and the classifier-free guidance scale $\gamma$. We find $T = 500$ or 1000 with $\gamma = 1.0$ or 2.0 consistently works well for both molecules and crystals. Additional ablation studies comparing joint vs. dataset-specific training, architecture variants, regularization techniques, and inference settings are presented in Appendix D.

**Evaluation metrics.** We evaluate the ability of ADiTs to sample valid and realistic molecules and crystals. Following prior work (Xie et al., 2022; Hoogeboom et al., 2022), we sample 10,000 crystals and molecules each and compute validity, stability, uniqueness and novelty rates using density functional theory (DFT) for crystals as well as validity, uniqueness and Posebusters sanity checks (Buttenschoen et al., 2024) for molecules. Detailed descriptions of all evaluation metrics are provided in Appendix B.

**Baselines.** We compare ADiT trained jointly on both QM9 and MP20 to molecule-only and crystal-only ADiT variants, as well as state-of-the-art baselines for both datasets. For crystal generation on MP20, we compare to: (1) three equivariant diffusion and flow matching-based models operating on multi-modal product manifolds: CDVAE (Xie et al., 2022), DiffCSP (Jiao et al., 2023), and FlowMM (Miller et al., 2024); (2) UniMat (Yang et al., 2024), a non-equivariant diffusion model which learns symmetries

from data; (3) FlowLLM (Sriram et al., 2024), a two-stage framework which first finetunes the autoregressive Llama 2 language model on crystal structures (Touvron et al., 2023; Gruver et al., 2024), and then trains FlowMM with samples from the language model as the base distribution and MP20 as the target distribution.

For molecule generation on QM9, we compare to: (1) Equivariant Diffusion (Hoogeboom et al., 2022), a roto-translationally equivariant diffusion model operating on a multi-modal product manifold; (2) GeoLDM (Xu et al., 2023), an alternative latent diffusion model using Equivariant Diffusion in the latent space of a roto-translationally equivariant autoencoder; (3) Symphony (Daigavane et al., 2024), an equivariant and autoregressive generative model that iteratively builds a molecule atom-by-atom.

## 4. Results

**State-of-the-art crystals and molecule generation.** Results for crystal generation in Table 1 show that ADiTs generate high-quality crystals compared to baseline diffusion models, achieving improved performance across validity, stability, uniqueness, and novelty metrics for 10,000 sampled crystals, with significant gains for compositional validity due to a single diffusion process in the VAE latent space rather than joint continuous and categorical diffusion. For molecule generation, ADiTs achieve state-of-the-art performance on validity and uniqueness metrics across 10,000 sampled molecules, as shown in Table 2(a), while Posebusters sanity check metrics in Table 2(b) further confirm that ADiTs generate physically realistic molecular structures, matching or exceeding baseline models across measures like double bond flatness, reasonable internal energy and lack of steric clashes.

Table 2: **Molecule generation results on QM9.** We report (a) validity and uniqueness rates, as well as (b) % pass rates on 7 sanity checks from Posebusters for 10,000 sampled molecules. ADiTs match or improve performance w.r.t. baselines, and sample physically realistic structures. Joint training with both molecular and crystal data improves molecular generation performance over QM9-only models. (* denotes models which explicitly generate hydrogen atoms.)

(a) **Validity results**

| | Model | Validity (%) ↑ | Unique (%) ↑ |
|---|---|---|---|
| QM9-only | Equivariant Diffusion | 97.50 | 96.71 |
| | Equivariant Diffusion* | 91.90 | 98.69 |
| | GeoLDM* | 93.80 | 98.82 |
| | Symphony* | 83.50 | 97.98 |
| | QM9-only ADiT | 96.02 | 97.76 |
| | QM9-only ADiT* | 92.19 | 97.90 |
| | Jointly trained ADiT | 97.43 | 96.92 |
| | Jointly trained ADiT* | 94.45 | 97.82 |

(b) **PoseBusters results**

| Test (% pass) ↑ | Symphony | Eq. Diff. | ADiT |
|---|---|---|---|
| Atoms connected | 99.92 | 99.88 | 99.70 |
| Bond angles | 99.56 | 99.98 | 99.85 |
| Bond lengths | 98.72 | 100.00 | 99.41 |
| Ring flat | 100.00 | 100.00 | 100.00 |
| Double bond flat | 99.07 | 98.58 | 99.98 |
| Internal energy | 95.65 | 94.88 | 95.86 |
| No steric clash | 98.16 | 99.79 | 99.79 |

(a) Crystals – MP20

(b) Molecules – QM9

Figure 2: **ADiTs are significantly faster than equivariant diffusion models.** We plot the number of integration steps for ADiTs and equivariant diffusion models vs. time to generate 10,000 samples on a single V100 GPU. ADiTs scale significantly better with the number of integration steps compared to equivariant diffusion.

**Joint training improves performance.** Table 1 and Table 2 also show that jointly trained ADiTs (trained on both QM9 and MP20 together) exceed the performance of the MP20-only or QM9-only ADiTs for materials or molecules, respectively. Joint training improves validity and stability rates for both crystals and molecules, demonstrating effective transfer learning between periodic and non-periodic atomic systems. These results validate that ADiTs can effectively model diverse types of atomic systems within a single architecture.

**Scaling up ADiT denoiser improves performance.** In Figure 3, we see that generative modelling performance predictably improves as we scale the DiT denoiser from parameter counts of 32M (DiT-S) to 130M (DiT-B) all the way to 450M (DiT-L), even with our current modest dataset size of ~130K total samples. The diffusion training loss and validity rates consistently improve with larger model sizes, showing a clear benefit from scale. Strong correla-

tions between model size and performance metrics suggest further gains are possible from scaling both model size and data – Alexandria (2M inorganic crystals), ZINC (250M molecules), and the Protein Data Bank (200K biomolecular complexes) present promising opportunities for dataset scaling.

**Speedup compared to equivariant diffusion.** ADiTs achieve significant inference speedup compared to equivariant diffusion under the same hardware conditions, as shown in Figure 2. When generating 10,000 samples on a V100 GPU, ADiTs based on standard Transformers leads to better scaling with integration steps compared to FlowMM (Miller et al., 2024) for crystals and GeoLDM (Xu et al., 2023) for molecules, both of which use computationally intensive equivariant networks as denoisers. It is significantly more practical to scale up Transformers than equivariant networks, as seen by the faster inference speed of ADiT-B compared to 100× smaller equivariant baselines.

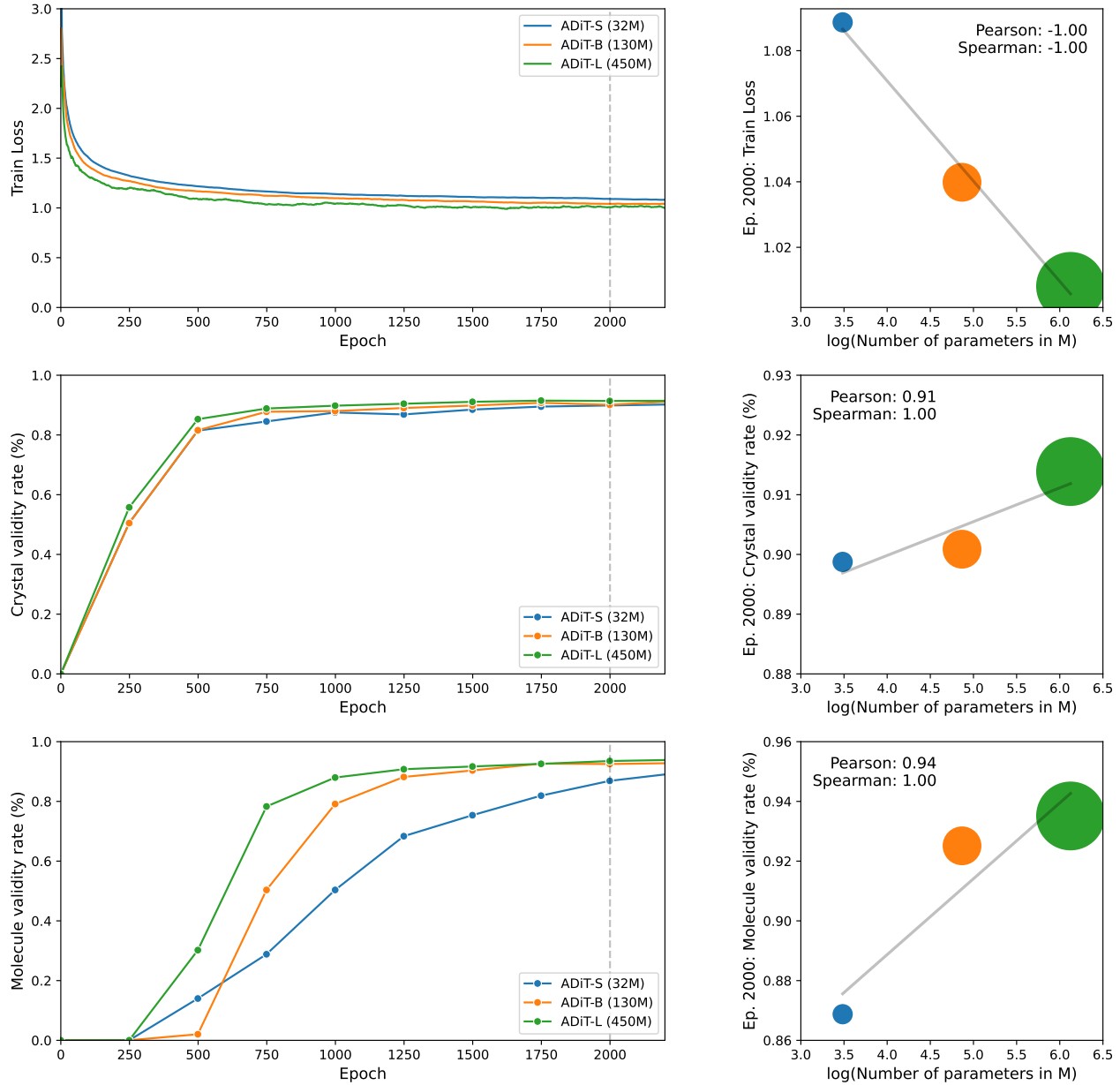

Figure 3: **Scaling up ADiT improves performance.** We show the effect of increasing the number of ADiT denoiser parameters on the training loss and generation validity rates. *Left:* training loss and validity rates vs. epochs. *Right:* Correlation plots for training loss and validity rates at epoch 2,000 vs. ADiT parameters (in Millions).

**Extension to larger GEOM-DRUGS molecules.** To demonstrate the scalability of the ADiT architecture to larger systems, we experiment with the GEOM-DRUGS dataset of 430,000 molecules of up to 180 atoms. Our setup follows Vignac et al. (2023) and we compare to state-of-the-art equivariant diffusion (Le et al., 2024) and flow matching (Irwin et al., 2025) baselines. In Table 3, ADiT is on par with equivariant models across validity and PoseBusters metrics. This is notable because ADiT is based on the standard Transformer architecture with minimal molecular inductive biases and, unlike equivariant baselines, does not explicitly predict atomic bonds.

**Additional results and visualizations.** For additional results and ablation studies, including extensions to metal-organic framework (MOF) generation, see Appendix C and Appendix D, respectively. In Appendix E, we further analyze ADiT's joint latent space via PCA and visualize some sampled crystals, molecules, and MOFs.

Table 3: **Molecule generation results on GEOM-DRUGS.** *Left:* Validity, uniqueness and % pass rates on Posebusters for 10,000 sampled molecules (* PoseBusters results taken from Buttenschoen et al. (2025)). ADiT with minimal molecular inductive biases matches or exceeds state-of-the-art equivariant diffusion baselines, which explicitly predict atomic bonds. *Right:* We plot the number of integration steps for ADiTs and SemlaFlow vs. time to generate 10,000 molecules on a single A100 GPU. ADiTs scale favorably with the number of integration steps compared to SemlaFlow, a highly optimized equivariant diffusion model.

| Metric (% pass) ↑ | EQGAT-diff* | SemlaFlow* | ADiT |
|---|---|---|---|
| Validity | 94.6 | 93.9 | 95.3 |
| Uniqueness | 100.0 | 100.0 | 100.0 |
| Atoms connected | 84.4 | 92.3 | 93.0 |
| Bond angles | 86.9 | 94.8 | 92.3 |
| Bond lengths | 87.0 | 94.6 | 92.5 |
| Ring flat | 87.0 | 94.9 | 95.4 |
| Double bond flat | 87.0 | 94.2 | 95.3 |
| Internal energy | 86.8 | 94.8 | 91.3 |
| No steric clash | 82.9 | 92.0 | 91.8 |
| PoseBusters valid | 59.7 | 87.5 | 85.3 |

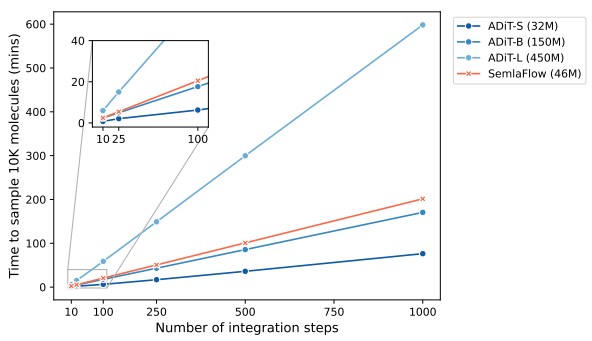

# 5. Discussions

Our work represents a significant step towards a broadly applicable foundation model for generative chemistry. We have introduced a unified latent diffusion framework for generating molecules and materials using a single model, and demonstrated the benefits of transfer learning from diverse atomic systems. The All-atom Diffusion Transformer (ADiT) obtains state-of-the-art performance in molecular generation using an architecture based primarily on standard Transformers with minimal inductive biases. This makes ADiT conceptually simpler and computationally more efficient than previous domain-specific approaches based on equivariant diffusion.

However, several limitations point to promising future directions. First, we currently use relatively small datasets for training, which may limit model generalization. Scaling to larger and more diverse datasets such as Alexandria and the Cambridge Structural Database for crystals, ZINC for small molecules, and the Protein Data Bank for biomolecular complexes could significantly improve performance and enable learning of broadly applicable chemical principles. Second, while we demonstrate success on small molecules and crystals of up to hundreds of atoms, we have not yet fully validated our approach on larger systems such as metal-organic frameworks or biomolecules containing thousands of atoms, though initial results for MOF generation are promising. Recent work on biomolecular structure prediction with AlphaFold3 (Abramson et al., 2024) demonstrates that simple Gaussian diffusion models with standard Transformers can effectively handle systems with thousands of atoms. Adapting ADiT to larger scales, while maintaining its unified representation across periodic and non-periodic

systems, could enable powerful transfer learning capabilities – especially valuable for low-data domains. Finally, our current models only perform unconditional generation – extending to conditional generation based on experimental properties, motif scaffolding, or molecular infilling would enable practical inverse design applications in drug discovery, materials science, and beyond.

# Impact Statement

Our work represents a step towards foundation models for chemistry, which could have significant benefits and risks for society. Our architecture unifies generative modelling of molecules and materials, enabling transfer learning across diverse atomic systems. Future AI systems for inverse design based on our architecture could accelerate the discovery of new materials and drugs, with potential applications in clean energy, drug discovery, and other areas critical to human health and sustainability. However, our methods could also be misused to design harmful molecular systems. We emphasize the importance of developing safeguards and guidelines for responsible deployment of these tools, and encourage the research community to carefully consider both positive and negative societal impacts as foundation models for chemistry continue to advance.

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

# A. Related Work

**Generative models for molecules and materials.** Diffusion models have emerged as the state-of-the-art for generative modelling of atomic systems, with applications to molecules, crystals, and biomolecules. For molecule generation, Equivariant Diffusion (Hoogeboom et al., 2022) pioneered roto-translationally equivariant diffusion on the multi-modal product manifold of atom types and 3D positions, while GeoLDM (Xu et al., 2023) introduced latent diffusion in the space of an equivariant autoencoder. Schneuing et al. (2024) extended equivariant diffusion to generate molecules conditioned on binding protein partners for structure-based drug design, while Corso et al. (2023) explored similar architectures for protein-small molecule docking. For crystal generation, state-of-the-art approaches use equivariant diffusion on product manifolds of atom types, 3D/fractional coordinates, and lattice parameters. Notable examples include CDVAE (Xie et al., 2022), DiffCSP (Jiao et al., 2023), and FlowMM (Miller et al., 2024). MatterGen (Zeni et al., 2025) demonstrated conditional diffusion for inverse design based on target material properties and symmetry space groups. Language models have also been used for generating molecules and crystals as textual representations (Flam-Shepherd & Aspuru-Guzik, 2023; Gruver et al., 2024).

Our work stands out as the first to develop unified generative models capable of sampling both periodic crystals and non-periodic molecular systems jointly. The closest work to ADiT in terms of diffusion formulation is AlphaFold3 (Abramson et al., 2024), which applies standard Transformers and Gaussian diffusion to generate all-atom biomolecular complex. However, their formulation is specific to structure prediction for biomolecules and only diffuses 3D atomic coordinates in Cartesian space. In contrast, our latent diffusion formulation is sufficiently general to work with both periodic and non-periodic systems, generating atom types, coordinates, as well as unit cell parameters unconditionally or with classifier-free guidance. Our emphasis on joint representations of molecules and crystals also aligns with recent work on general-purpose foundation models for molecular dynamics (Shoghi et al., 2024; Batatia et al., 2023). Similarly, our unified latent diffusion framework can potentially be scaled up with larger and more diverse chemical datasets towards foundation models for generative chemistry.

**Latent diffusion models.** Latent diffusion models (Vahdat et al., 2021; Rombach et al., 2022) propose to do diffusion in the latent space of an autoencoder instead of the raw input space of high-dimensional continuous signals such as pixels, and have been extremely successful for generating images, audio, and videos (Esser et al., 2024; Betker et al., 2023; Brooks et al., 2024). Latent diffusion is a more computationally efficient alternative to standard diffusion as the autoencoder's latent space captures semantically meaningful features of the data, allowing for more efficient diffusion in a lower-dimensional space followed by reconstruction to the original data space. The original formulation was further improved by Diffusion Transformers (DiTs) (Peebles & Xie, 2023), which demonstrated that standard Transformers provide a highly scalable architecture for the denoiser network. Latent diffusion models can easily incorporate conditioning on additional information like class labels, text prompts, or infilling masks through classifier-based (Dhariwal & Nichol, 2021) and classifier-free guidance (Ho & Salimans, 2022) as well as finetuning (Zhang et al., 2023; Dai et al., 2023).

Our work is the first to leverage latent diffusion for jointly generating the complex multi-modal product of categorical and continuous data types that constitute 3D atomic systems. This allows us to shift the complexity of handling atom types, coordinates, and unit cell parameters into an autoencoder while performing the generative process in latent space with DiTs, which is simpler and more scalable than alternative multi-modal equivariant diffusion models.

**Equivariance and generative modelling.** Geometric Graph Neural Networks (Duval et al., 2023), particularly roto-translationally equivariant networks, have been used as denoisers in diffusion and flow matching approaches for generative modeling of 3D atomic systems. E(3)-Equivariant Graph ConvNets (Satorras et al., 2021) are widely used as denoisers for molecule (Hoogeboom et al., 2022; Xu et al., 2023; Schneuing et al., 2024) and crystal generation (Jiao et al., 2023; Miller et al., 2024). More expressive architectures, like higher-order tensor networks (Liao et al., 2024) and Invariant Point Attention (Jumper et al., 2021), have been applied to protein structure generation (Watson et al., 2023; Yim et al., 2023b) and protein-ligand docking (Corso et al., 2023).

However, equivariant networks are computationally expensive and harder to scale than standard Transformers in terms of data and model size. This is especially relevant for diffusion models, where denoisers typically process inputs as fully connected graphs to capture global structure (Joshi, 2020) and are iteratively run hundreds of times during inference. Recent work has challenges the necessity of 3D inductive biases and equivariance for generative structure prediction tasks, showing that standard Transformers can achieve strong performance on biomolecular complexes (Abramson et al., 2024) and small molecule conformations (Wang et al., 2024; O Pinheiro et al., 2023). Non-equivariant models have also shown promising results for protein structure generation (Chu et al., 2024; Martinkus et al., 2024). In the same vein, our work leverages

the simplicity and scalability of standard Transformers for generative modelling across both periodic and non-periodic 3D atomic systems, demonstrating that explicit equivariance and molecular inductive biases are not a strict requirement for generating valid and realistic atomic structures at scale.

## B. Evaluation Metrics

**Crystal generation metrics.** We follow the evaluation protocol established by Xie et al. (2022); Miller et al. (2024), where we sample 10,000 crystals and compute validity, stability, uniqueness, and novelty rates, defined as follows:

- Structural validity: % of crystals with all pairwise distances $>= 0.5$ and crystal volume $>= 0.1$.
- Compositional validity: % of crystal compositions with charge neutrality and electronegativity balance according to SMACT (Davies et al., 2019).
- Overall validity: % of crystals which are both structurally and compositionally valid.
- Stability: % of crystals with DFT energy above hull $<0.0$ eV/atom and no. of unique elements $>= 2$. (We also report metastability as DFT energy above hull $<0.1$ eV/atom and no. of unique elements $>= 2$.)
- Stable & unique: % of stable crystals which are unique, as defined by an all-to-all comparison using Structure Matcher[2] from PyMatGen (Ong et al., 2013).
- Stable, unique & novel: % of stable, unique crystals which are novel, as defined by an all-to-all comparison to all crystals in MP-20 using Structure Matcher.

To compute the stability, uniqueness, and novelty rates, we follow Miller et al. (2024); Sriram et al. (2024): We first pre-relax the sampled crystals using a fast ML potential, CHGnet (Deng et al., 2023), and then perform DFT relaxation. We then determine the DFT energy above hull for the relaxed structures against the Matbench Discovery convex hull (Riebesell et al., 2023). Note that there is a lower bound on the number of completed DFT calculations due to memory or timeout errors.

**Molecule generation metrics.** We follow the evaluation protocol established by Hoogeboom et al. (2022); Daigavane et al. (2024), where we sample 10,000 molecules and compute validity and uniqueness rates as well as success rates for 7 sanity checks from Posebusters (Buttenschoen et al., 2024), as follows:

- Validity: % of molecules with canonical SMILES string found by RDKit.
- Uniqueness: % of unique SMILES among valid ones.
- All-atoms connected: % of molecules where there exists a path along bonds between all atoms.
- Reasonable bond angles/lengths: % of molecules where all angles/lengths are within 0.75 of the lower and 1.25 of the upper bounds determined by distance geometry.
- Aromatic rings flatness: % of molecules where All-atoms in aromatic rings with 5 or 6 members are within 0.25Å of the closest shared plane molecule.
- Double bond flatness: % of molecules where All-atoms of aliphatic carbon-carbon double bonds and their four neighbours are within 0.25Å of the closest shared plane.
- Reasonable internal energy: % of molecules where the calculated energy is no more than 100 times the average energy of an ensemble of 50 conformations generated for the input molecule.
- No internal steric clash: % of molecules where the interatomic distance between pairs of non-covalently bound atoms is above 0.8 of the distance geometry lower bound.

The validity and uniqueness metrics focus on whether the chemical composition of generated molecules can be processed by RDKit, while the Posebusters sanity checks evaluate the physical realism of the generated 3D structures across multiple criteria, from geometric constraints like bond lengths to energetic considerations (Harris et al., 2023).

## C. Additional Results

**Extension to MOF generation.** Having established that ADiTs benefit from transfer learning to generate high-quality crystals and molecules, we challenged our architecture to generate metal-organic frameworks (MOFs), which represent a more complex class of hybrid materials with metal nodes connected by organic small molecule linkers. We trained ADiTs on an additional 14,000 MOFs of up to 150 atoms from the QMOF database (Rosen et al., 2021) alongside QM9 and MP20,

---

[2]Structure Matcher checks if two periodic structures are equivalent, even if they are in different settings or have minor distortions.

using the same experimental settings and training for 10,000 epochs. We sampled 1,000 MOFs and evaluated their validity using 15 sanity checks from MOFChecker (Jablonka, 2023), including tests for: presence of metal/carbon/hydrogen atoms, atomic overlaps, over/undervalent carbons and nitrogens, missing hydrogens, and excessive partial charges.

Table 4 shows that QMOF-only trained ADiT achieves a 15% overall validity rate for MOF generation, which decreases to 10% with joint training on molecules, crystals, and MOFs simultaneously. However, the jointly trained model takes significant time to train and did not fully converge, suggesting that further improvements in MOF generation may be possible with larger models trained for longer. Comparing validity rates for joint vs. dataset-specific ADiTs shows that the joint model benefits from transfer learning across molecules, crystals, and MOFs, achieving high validity rates earlier in training. Notably, the joint model achieves high validity rates of 91% for crystals and 95% for molecules, matching our best dataset-specific models and being able to additionally generate MOFs. While not directly comparable, models specialized for MOF such as MOFDiff (Fu et al., 2024) are trained on large synthetic dataset of 300,000 MOFs and achieve 30% validity rates based on MOFChecker *after* DFT relaxation of generated MOFs. Our results are reported without DFT relaxation.

Table 4: **Metal-organic framework generation results.** *Left:* We report sanity checks from MOFChecker for 1,000 sampled MOFs, trained on QMOF-only as well as jointly with QM9 and MP20. (↑/↓ indicate higher/lower is better, respectively.) *Right:* Jointly trained and dataset-specific ADiT validity rates vs. epochs. The joint model benefits from transfer learning and requires fewer epochs per dataset to start generating valid samples.

| Test (%) | QMOF ADiT | Joint ADiT |
|---|---|---|
| Has carbon ↑ | 100.0 | 100.0 |
| Has hydrogen ↑ | 99.6 | 100.0 |
| Has atomic overlap ↓ | 8.3 | 10.8 |
| Has overcoord. C ↓ | 23.6 | 34.3 |
| Has overcoord. N ↓ | 1.5 | 1.6 |
| Has overcoord. H ↓ | 1.0 | 3.6 |
| Has undercoord. C ↓ | 60.0 | 72.1 |
| Has undercoord. N ↓ | 39.1 | 39.9 |
| Has undercoord. rare earth ↓ | 0.4 | 0.8 |
| Has metal ↑ | 100.0 | 99.4 |
| Has lone molecule ↓ | 72.9 | 83.2 |
| Has high charge ↓ | 0.9 | 2.5 |
| Has suspicious terminal oxo ↓ | 2.6 | 5.8 |
| Has undercoord. alkali ↓ | 1.0 | 6.4 |
| Has geom. exposed metal ↓ | 7.0 | 9.6 |
| Validity rate (all passed) ↑ | 15.7 | 10.2 |

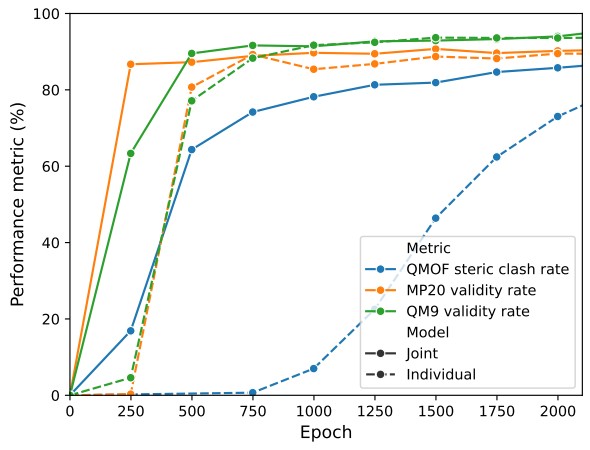

**Histograms from DFT validation.** In Figure 4, we show histograms of DFT energy above hull, formation energy, and number of unique elements per crystal for 10,000 generated crystals from ADiT, FlowMM, and FlowLLM compared to the MP20 training distribution. ADiT generates more thermodynamically stable crystals than prior models, as shown by the larger proportion of samples with DFT energy above hull below 0.0 eV/atom. The distribution of DFT formation energies and number of unique elements per crystal from ADiT samples more closely matches the MP20 training data compared to FlowMM and FlowLLM baselines, suggesting that ADiT better captures the underlying physical and chemical constraints of stable crystal structures. Note that we ran DFT calculations for all model samples under identical hardware and settings to ensure fair comparison.

**Histogram of spacegroups.** In Figure 5, we show the distribution of spacegroups for 10,000 generated crystals from ADiT, FlowMM, FlowLLM and the MP20 distribution. Diffusion-based models (ADiT and FlowMM) tend to over sample crystals with P1 spacegroup, which represents the lowest symmetry group, likely due to their local, step-wise denoising process. In contrast, FlowLLM, an autoregressive language model, tends to over sample spacegroups like Fm-3m, Pm-3m, and I4/mmm compared to the training data. While it would be straightforward to control the distribution of spacegroups generated by ADiT through classifier-free guidance conditioning, we leave this for future work since our current focus is on unconditional generation of diverse atomic systems.

**SUN rate and scaling ADiT.** In Table 5, we observe that the combined stability, uniqueness, and novelty (S.U.N.) rate for crystal generation decreases as we scale up the DiT denoiser from DiT-S (32M) to DiT-L (450M). While stability and

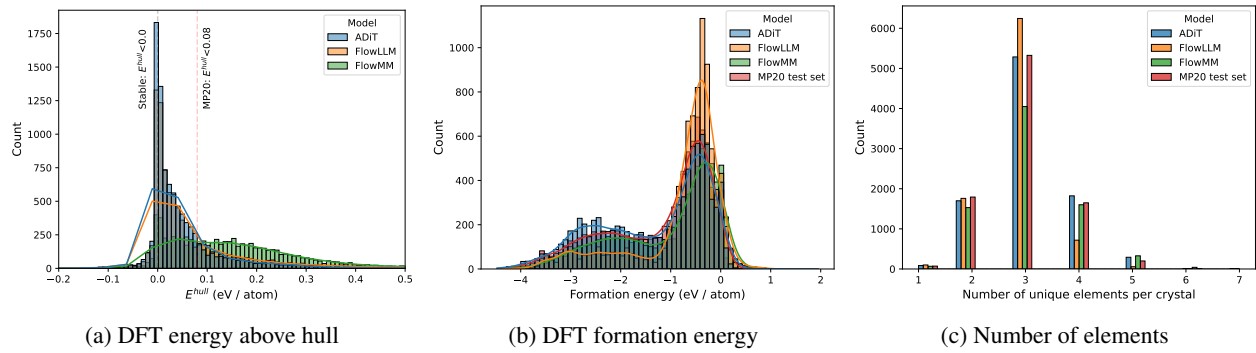

(a) DFT energy above hull      (b) DFT formation energy      (c) Number of elements

Figure 4: **Histograms from DFT validation of 10,000 generated crystals.** ADiT is more likely to generate stable crystals with DFT energy above hull <0.0 eV/atom compared to prior models. Samples from ADiT most closely follow the distributions for DFT formation energy and number of unique elements per crystal from MP20.

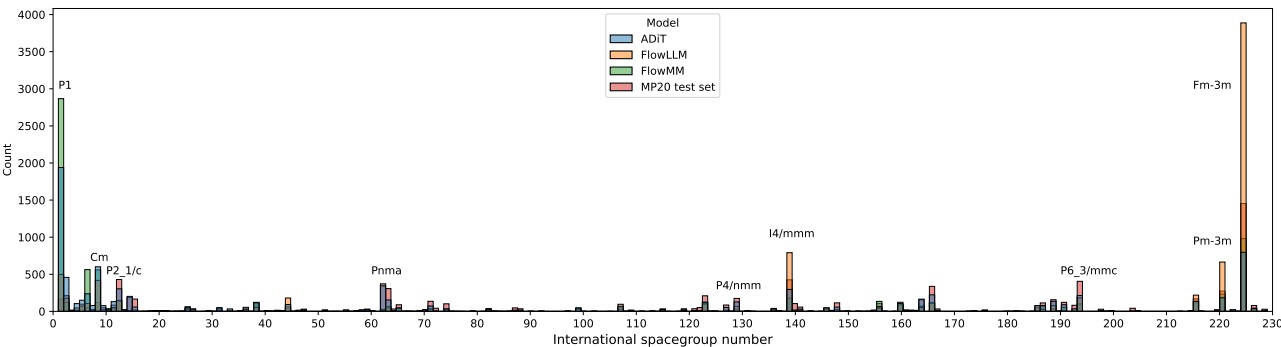

Figure 5: **Histogram of spacegroups for 10,000 generated crystals.** Diffusion-based ADiT and FlowMM tend to over sample crystals with P1 spacegroup compared to the MP20 training distribution. FlowLLM, an autoregressive language, tends to over sample crystals with Fm-3m, Pm-3m, and I4/mmm spacegroups.

uniqueness rates increase with model size, the S.U.N. rate decreases due to the larger model's greater capacity to memorize the small MP20 training dataset of 27K crystals. This suggests that larger models may be more prone to generating duplicate or near-duplicate samples, which we plan to address by training on larger and more diverse datasets in future work. For crystals, the Alexandria dataset of inorganic crystals and the Crystallography Open Database of organic crystals present promising opportunities for scaling up. Notably, ADiT-S trained on MP20-only achieves a S.U.N. rate of 6.5%, representing a significant improvement over previously published results from FlowMM (2.8%) and FlowLLM (4.7%). This demonstrates that even our smallest model variant substantially advances the state-of-the-art for crystal generation.

**Sensitivity of validity rate to number of samples and random seed.** In Figure 6a, we plot the validity rates for crystal and molecule generation as we increase the number of samples from 100 to 10,000 for 3 different random seeds. We observe that the validity rates generally converge and are stable across random seeds after sampling over 5,000 crystals or molecules.

**Sensitivity of S.U.N. rate to number of samples.** In Figure 6b, we plot the S.U.N. (stability, uniqueness, and novelty) rates for crystal generation as we increase the number of samples from 100 to 10,000 across 3 different random seeds. The S.U.N. rates converge after approximately 5,000 samples for diffusion-based methods like ADiT and FlowMM. In contrast, autoregressive models like FlowLLM show higher variance in S.U.N. rates, likely due to more frequent generation of duplicate crystals during low-temperature sampling.

Table 5: **Impact of scaling on stability, uniqueness, and novelty rates for 10,000 generated crystals.** We find that stability rate as well as stability & uniqueness rate increase as we increase the number of model parameters for ADiT from 32M to 450M. However, larger ADiT models have greater capacity to memorise the small MP20 training dataset of 27K crystals, resulting in decrease in the combined stability, uniqueness, & novelty rate. ADiT-S trained on MP20-only achieves a S.U.N. rate of 6.5%, representing a significant improvement over previously published state-of-the-art models which attained S.U.N. rates up to 4.7%.

| Model | Stability ($E^{hull} < 0.0$) | | | Metatability ($E^{hull} < 0.1$) | | |
|---|---|---|---|---|---|---|
| | S (%) ↑ | S.U. (%) ↑ | S.U.N. (%) ↑ | M.S (%) ↑ | M.S.U. (%) ↑ | M.S.U.N. (%) ↑ |
| MP20-only ADiT-S (32M) | 12.8 | 11.8 | 6.5 | 71.1 | 64.9 | 38.1 |
| MP20-only ADiT-B (130M) | 14.1 | 12.5 | 4.7 | 81.6 | 67.3 | 25.9 |
| Joint ADiT-S (32M) | 12.6 | 11.4 | 6.0 | 71.9 | 64.7 | 37.7 |
| Joint ADiT-B (130M) | 15.4 | 13.4 | 5.3 | 81.0 | 70.2 | 28.2 |
| Joint ADiT-L (450M) | 15.5 | 13.5 | 5.0 | 82.5 | 70.9 | 27.9 |

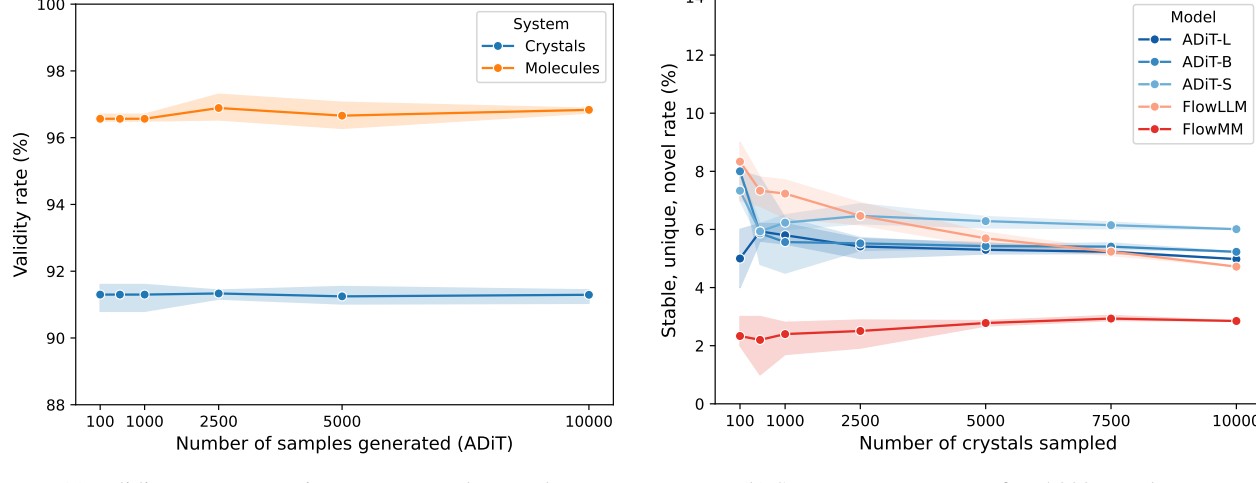

(a) Validity rates are consistent across random seeds.

(b) S.U.N. rates converge after 5,000 samples.

Figure 6: **Consistency of validity and S.U.N. rates as we increase number of samples.** We plot the validity and S.U.N. rates vs. number of sampled crystals or molecules. Error bars indicate 95% confidence interval across three different random seeds. Metrics are generally stable across seeds and converge after sampling over 5,000 crystals or molecules.

# D. Ablation Study

Table 7 and Table 6 presents ablation studies as well as aggregated benchmarks for various configurations of ADiT's latent diffusion model and autoencoder, respectively. Key takeaways are highlighted below. Note that, unless otherwise stated, results in the main paper are reported for jointly trained ADiT-B which uses DiT-B denoiser, standard Transformer encoder and decoder, latent dimension $d = 8$, and KL regularization weight $\lambda_{\text{KL}} = 1e - 5$.

**Joint vs. dataset-specific training**  Joint training of the autoencoder to embed both molecules and crystals into a shared latent space achieves similar or better reconstruction performance compared to dataset-specific training, as shown in Table 6 (rows 3, 6, 10). The benefits of joint training are most evident in generative modelling performance – samples from the joint model have higher validity rates for both crystals and molecules compared to dataset-specific models, demonstrating effective transfer learning between periodic and non-periodic atomic systems (Table 7, rows 12, 16, 20). These results provide strong evidence that ADiTs can successfully unify the modelling of both periodic and non-periodic atomic systems within a single architecture, without compromising performance on either domain.

**Denoiser architecture**  The DiT denoiser is a standard Transformer with key hyperparameters including the hidden dimension $d_{\text{model}}$, number of attention heads, and number of layers. Scaling up the DiT denoiser from DiT-S (32M parameters, $d_{\text{model}} = 384$, 6 heads, 12 layers) to DiT-B (150M, $d_{\text{model}} = 768$, 12 heads, 12 layers) and DiT-L (450M, $d_{\text{model}} = 1024$, 24 heads, 24 layers) consistently improves generative performance, as shown in Table 7 (rows 12, 16, 20). We have additionally performed scaling law analysis for the training loss and validity rates in Figure 3, seeing strong correlations between model size and performance metrics. In Figure 6b, we further see that S.U.N. rates for larger models are better than smaller models, further confirming the benefits of scaling up the DiT denoiser.

**Autoencoder architecture**  For the architecture of the autoencoder's encoder and decoder, we explored both roto-translation equivariant as well as non-equivariant VAEs. For the equivariant VAE variant, the encoder is Equiformer-V2 (Liao et al., 2024) and the decoder is an equivariant feedforward network adapted from output heads in the Equiformer-V2 codebase. We selected Equiformer-V2 as it is theoretically expressive (Joshi et al., 2023) and has state-of-the-art performance across diverse 3D atomic systems. As input to the Equiformer-V2 encoder, we use spherical harmonic embeddings of displacement vectors as edge features and exclude the 3D coordinates in Algorithm 1, line 2, from the initial features $\{h_i\}$ as a result. The initial features $\{h_i\}$ are used as the $L = 0$ scalar component of the initial spherical tensor features of Equiformer-V2. The rest of the pseudocode in Algorithms 1 and 2 remains the same.

As shown in Table 6 (rows 1-4 and 5-8), the choice of autoencoder architecture has noticeable impact on reconstruction performance. Standard Transformers generally outperform Equiformer-V2 for both crystals and molecules, achieving higher match rates (% of test set samples where the reconstructed structure matches the groundtruth, as determined by PyMatGen's StructureMatcher/MoleculeMatcher). More importantly, the latent space learned by standard Transformers proved more suitable for the latent diffusion process compared to Equiformer-V2's equivariant latent space, leading to substantially better generative performance in terms of validity rates, particularly for crystals (Table 7, rows 1-4 and 5-8).

**Autoencoder regularization**  As shown in Table 6 (rows 9-12), increasing the latent dimension and reducing the KL regularization weight generally improved autoencoder reconstruction performance by lowering RMSD values which measure the average distance between the reconstructed and groundtruth structures. These improvements in reconstruction quality translated to better generative performance, with higher validity rates for both crystals and molecules at larger latent dimensions and lower KL weights (see Table 7, rows 9-12).

**Sampling hyperparameters.**  Classifier-free guidance scale and number of integration steps are important hyperparameters for inference-time tuning. In Figure 7, we show a grid search over guidance scales $\gamma \in \{1.0, 2.0, 3.0, 4.0, 6.0\}$ and integration steps $T \in \{10, 50, 100, 250, 500, 1000\}$, finding that different combinations may be optimal for crystals vs. molecule generation. For each entry in Table 7, we have reported results for $T$ and $\gamma$ which obtain the highest validity rates. $T = 500$ or $1000$ with $\gamma = 1.0$ or $2.0$ tends to work well across both molecules and crystals.

Table 6: **Autoencoder ablation study.** We report match rate (computed with StructureMatcher or MoleculeMatcher from PyMatGen) and RMSD between the reconstructed and groundtruth structures for MP20 crystals and QM9 molecules.

| Train Set | Autoencoder hyperparameters | | | Crystals – MP20 | | Molecules – QM9 | |
|---|---|---|---|---|---|---|---|
| | Encoder | Latent | KL | Match Rate (%) ↑ | RMSD (Å) ↓ | Match Rate (%) ↑ | RMSD (Å) ↓ |
| MP20 | Transformer | 4 | 0.0001 | 85.50 | 0.0598 | - | - |
| MP20 | Equiformer-V2 | 4 | 0.0001 | 81.70 | 0.1652 | - | - |
| MP20 | Transformer | 8 | 0.0001 | 84.50 | 0.0502 | - | - |
| MP20 | Equiformer-V2 | 8 | 0.0001 | 88.90 | 0.0296 | - | - |
| QM9 | Transformer | 4 | 0.0001 | - | - | 97.20 | 0.0747 |
| QM9 | Equiformer-V2 | 4 | 0.0001 | - | - | 96.20 | 0.0765 |
| QM9 | Transformer | 8 | 0.0001 | - | - | 96.50 | 0.0823 |
| QM9 | Equiformer-V2 | 8 | 0.0001 | - | - | 96.20 | 0.0746 |
| Joint | Transformer | 4 | 0.0001 | 88.30 | 0.0471 | 96.60 | 0.0785 |
| Joint | Transformer | 4 | 0.00001 | 88.50 | 0.0468 | 98.50 | 0.0524 |
| Joint | Transformer | 8 | 0.0001 | 88.60 | 0.0269 | 96.60 | 0.0760 |
| Joint | Transformer | 8 | 0.00001 | 88.60 | 0.0239 | 97.00 | 0.0399 |

Table 7: **Latent diffusion model ablation study.** We report validity rates for 10,000 generated crystals or molecules.

| Train Set | Diffusion Denoiser | Autoencoder hyperparameters | | | Crystals – MP20 | | | Molecules – QM9 | |
|---|---|---|---|---|---|---|---|---|---|
| | | Encoder | Latent | KL | Structure Valid (%) ↑ | Composition Valid (%) ↑ | Overall Valid (%) ↑ | Validity (%) ↑ | Validity* (%) ↑ |
| MP20 | DiT-S | Transformer | 4 | 0.0001 | 98.90 | 89.19 | 88.19 | - | - |
| MP20 | DiT-S | Equiformer-V2 | 4 | 0.0001 | 91.74 | 81.03 | 74.43 | | |
| MP20 | DiT-S | Transformer | 8 | 0.0001 | 99.58 | 90.46 | 90.13 | - | - |
| MP20 | DiT-S | Equiformer-V2 | 8 | 0.0001 | 99.26 | 86.09 | 85.50 | | |
| QM9 | DiT-S | Transformer | 4 | 0.0001 | - | - | - | 95.94 | 92.19 |
| QM9 | DiT-S | Equiformer-V2 | 4 | 0.0001 | - | - | - | 95.36 | 91.37 |
| QM9 | DiT-S | Transformer | 8 | 0.0001 | - | - | - | 96.02 | 91.58 |
| QM9 | DiT-S | Equiformer-V2 | 8 | 0.0001 | - | - | - | 96.24 | 91.47 |
| Joint | DiT-S | Transformer | 4 | 0.0001 | 98.21 | 91.05 | 89.38 | 96.90 | 93.47 |
| Joint | DiT-S | Transformer | 4 | 0.00001 | 98.74 | 90.74 | 89.60 | 96.40 | 91.85 |
| Joint | DiT-S | Transformer | 8 | 0.0001 | 99.66 | 91.07 | 90.76 | 96.85 | 93.33 |
| Joint | DiT-S | Transformer | 8 | 0.00001 | 99.67 | 91.25 | 90.93 | 96.36 | 92.06 |
| Joint | DiT-B | Transformer | 4 | 0.0001 | 99.00 | 91.23 | 90.29 | 97.33 | 94.45 |
| Joint | DiT-B | Transformer | 4 | 0.00001 | 99.51 | 90.73 | 90.29 | 97.04 | 94.06 |
| Joint | DiT-B | Transformer | 8 | 0.0001 | 99.67 | 91.60 | 91.32 | 95.30 | 89.85 |
| Joint | DiT-B | Transformer | 8 | 0.00001 | 99.74 | 92.14 | 91.92 | 97.43 | 93.99 |
| Joint | DiT-L | Transformer | 4 | 0.0001 | 99.31 | 90.92 | 90.29 | 97.80 | 94.67 |
| Joint | DiT-L | Transformer | 4 | 0.00001 | 99.43 | 90.84 | 90.31 | 96.71 | 92.78 |
| Joint | DiT-L | Transformer | 8 | 0.0001 | 99.75 | 92.17 | 91.92 | 96.11 | 91.45 |
| Joint | DiT-L | Transformer | 8 | 0.00001 | 99.66 | 91.42 | 91.14 | 97.79 | 95.01 |

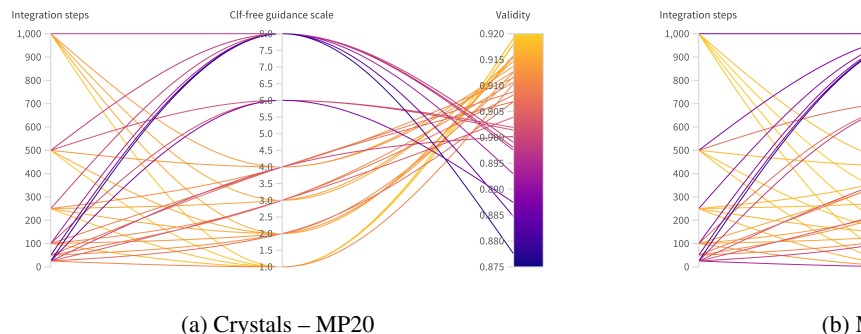

(a) Crystals – MP20                                    (b) Molecules – QM9

Figure 7: **Tuning inference hyperparameters for best performance.** Best generative modelling results for crystals and molecules are achieved with different classifier-free guidance scales $\gamma$ and number of integration steps $T$. $T = 500$ or $1000$ with $\gamma = 1.0$ or $2.0$ tends to work well across both molecules and crystals.

# E. Visualizations

**PCA visualization of shared latent space.** In Figure 8, we plot the first two PCA principal components of 100 random samples each from the MP20 and QM9 validation set, as well as 100 generated crystals and 100 generated molecules sampled from ADiT. We observe that the joint latent space shows distinct clusters between molecules and crystals, with tighter clustering for molecules and more spread for crystals, reflecting the greater diversity of elements and local geometric environments in periodic crystal structures.

Next, we plot the same PCA but only keeping atoms of carbon, nitrogen, oxygen, and fluorine in Figure 9. These atoms appear in both QM9 molecules and MP20 crystals, allowing us to analyze how their representations compare across periodic and non-periodic systems. The visualization reveals clear patterns: principal component 1 primarily distinguishes between molecules (clustered between -2 and 2) and crystals, while principal component 2 correlates with atom type. Most notably, oxygen atoms show similar latent representations whether they appear in molecules or crystals, suggesting ADiT's latent space captures fundamental chemical properties that transfer across both domains. This shared representation of oxygen, a key element in both datasets, may help explain ADiT's successful joint learning and transfer between periodic and non-periodic systems.

**Generated crystals, molecules, and MOFs.** In Figure 10, Figure 11, and Figure 12, we show samples of generated crystals, molecules, and MOFs from ADiT, respectively. The generated crystals exhibit diverse spacegroups and compositions, while the generated molecules show a wide range of chemical structures and conformations. These visualizations demonstrate that the jointly trained ADiT model successfully generates high-quality and chemically diverse atomic systems in both periodic and non-periodic domains.

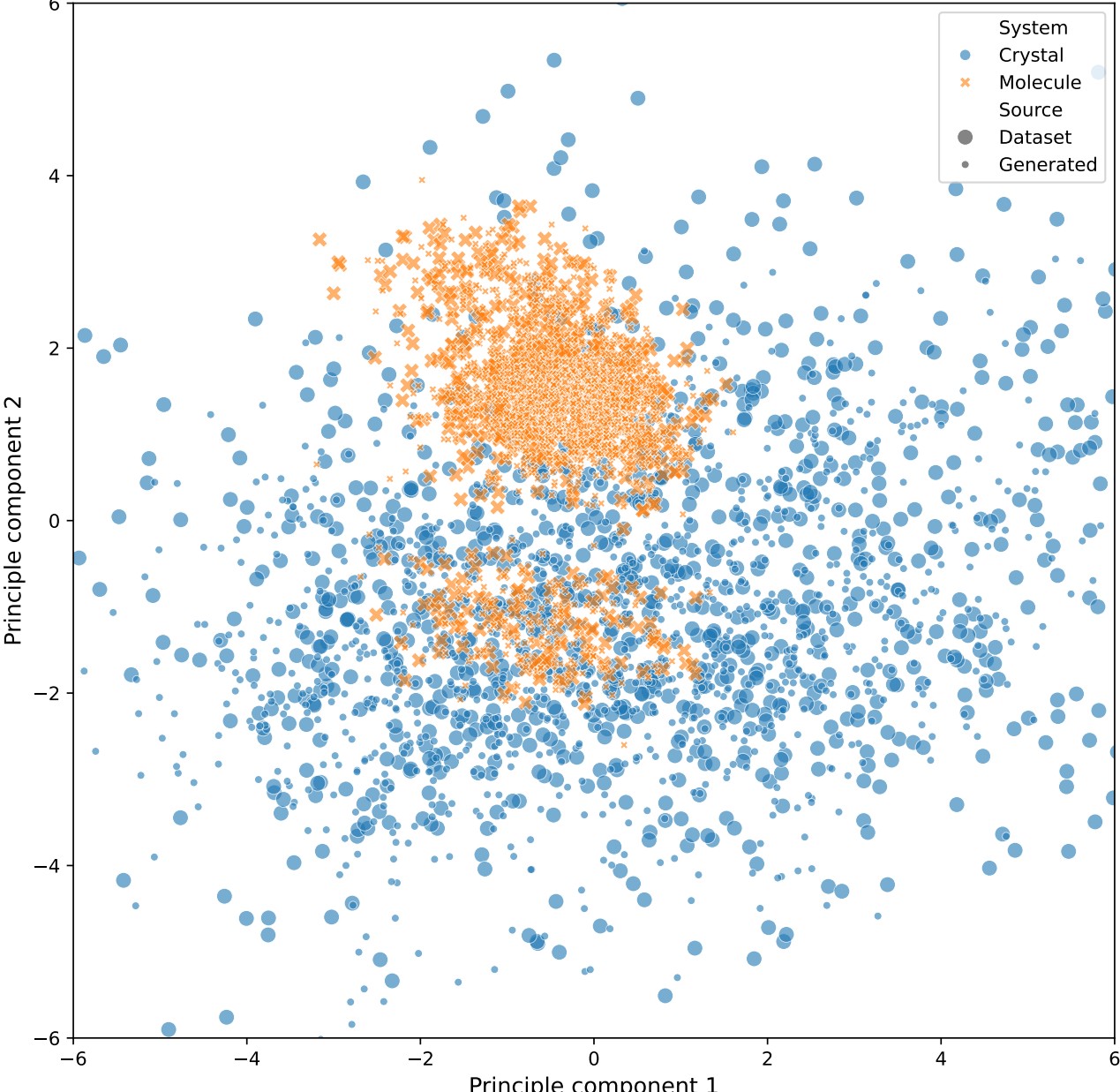

Figure 8: PCA plot of latent embeddings from ADiT's VAE for 100 data points from the MP20 and QM9 datasets, as well as 100 ADiT-generated crystals/molecules each. Each point represents an atom, coloured by the system type and sized by whether it comes from real data or generated latents. **The joint latent space shows distinct clusters between molecules and crystals, with tighter clustering for molecules and more spread for crystals, reflecting the greater diversity of elements and local geometric environments in periodic crystal structures.**

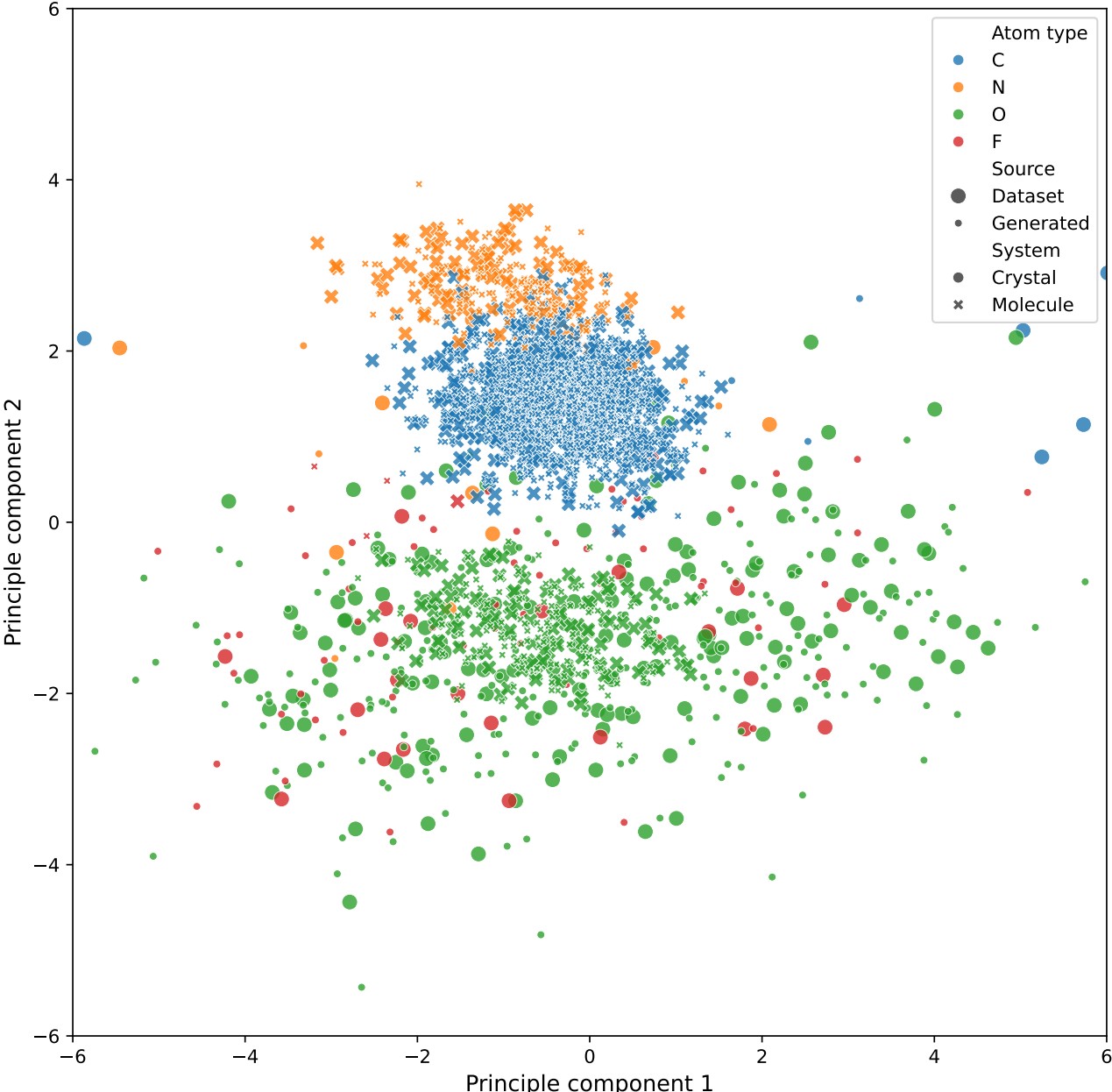

Figure 9: PCA plot of latent embeddings for carbon, nitrogen, oxygen, and fluorine atoms from ADiT's VAE for 100 data points from the MP20 and QM9 datasets, as well as 100 ADiT-generated crystals/molecules each. Each point represents an atom, coloured by atom type and sized by whether it comes from real data or generated latents. Principle component 1 visually correlates with whether a system is a molecule (within range -2 – 2) or crystal. Principle component 2 visually correlates with the atom type. **The joint latent space shows distinct clusters for different atom types, with oxygen atoms having similar representations in both molecules and crystals. This overlap in oxygen atom representations suggests that ADiT's latent space captures shared chemical properties across periodic and non-periodic systems, enabling effective knowledge transfer during joint training.**

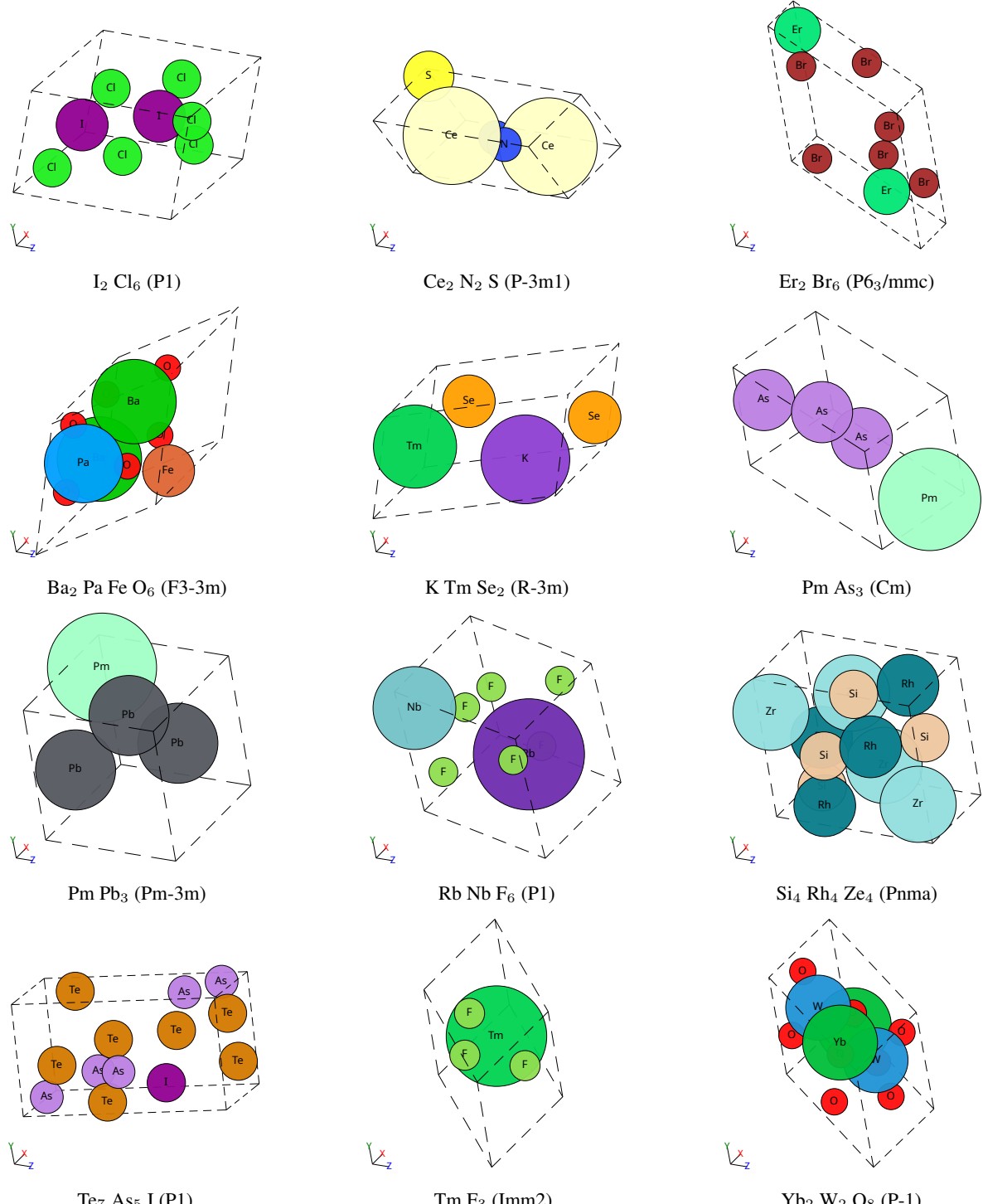

Figure 10: Generated crystals from ADiT trained jointly on MP20 crystals and QM9 molecules.

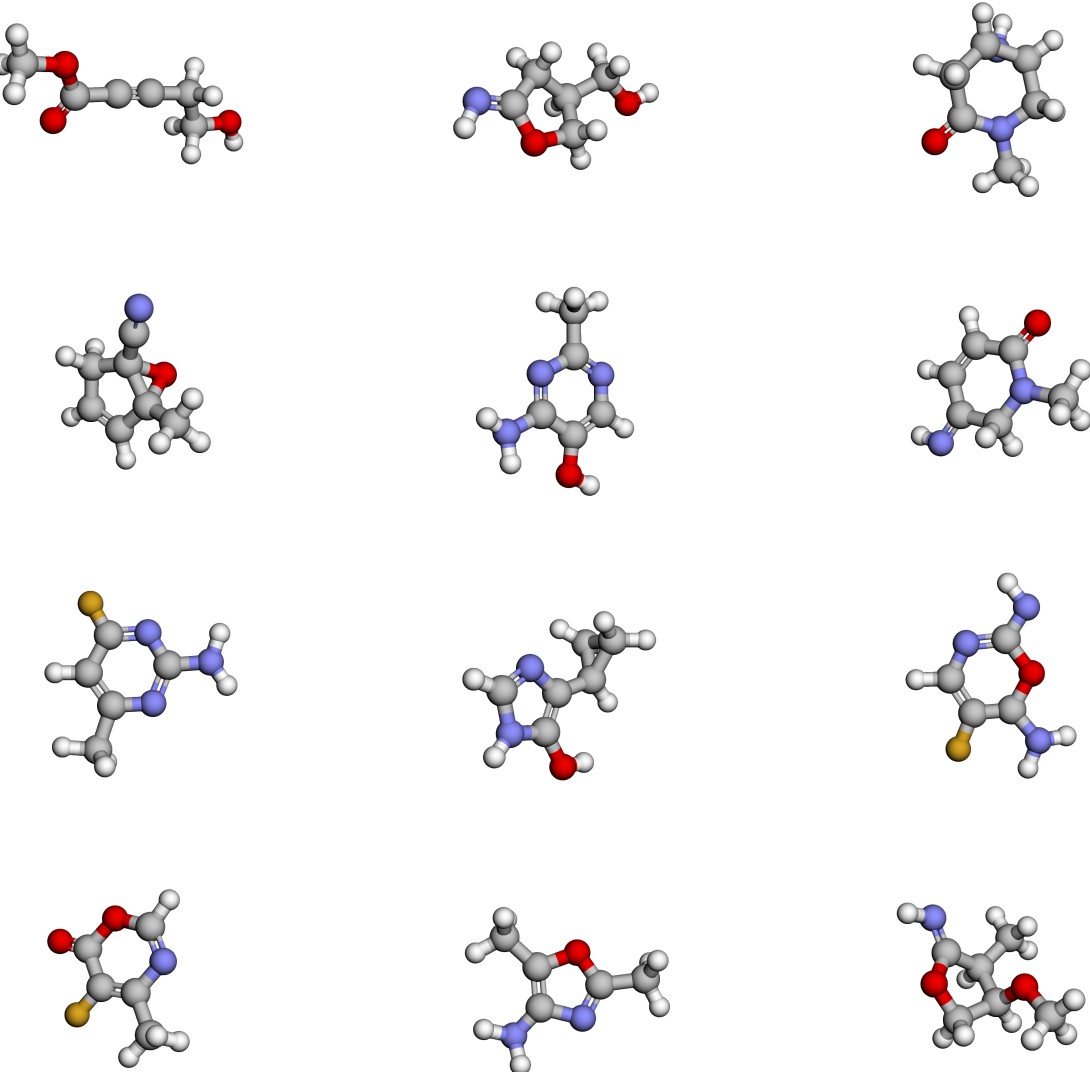

Figure 11: Generated molecules from ADiT trained jointly on MP20 crystals and QM9 molecules.

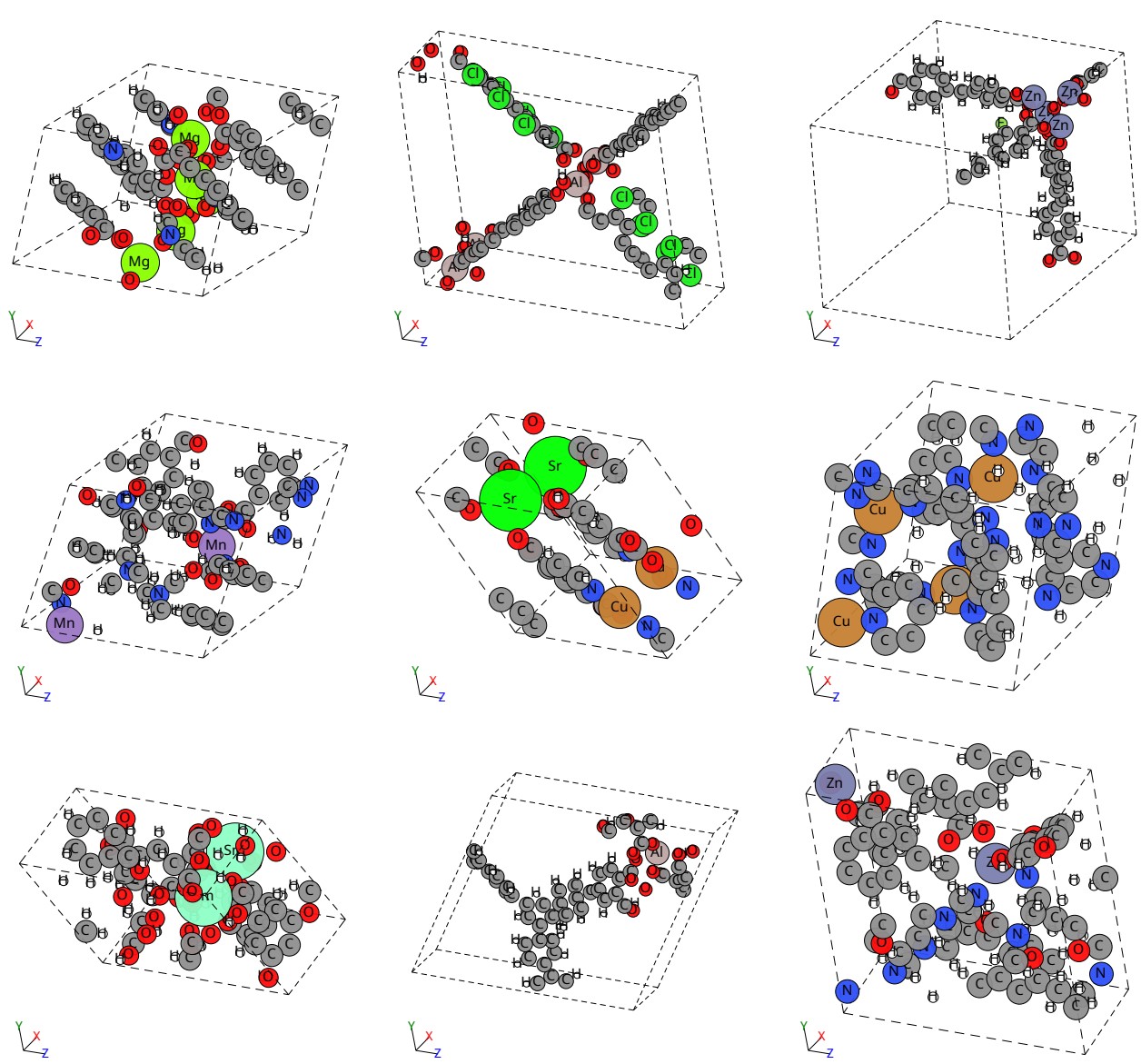

Figure 12: Generated metal-organic frameworks from ADiT trained jointly on QMOF150 metal-organic frameworks, MP20 crystals and QM9 molecules.

