# OpenReview forum: "All-atom Diffusion Transformers: Unified generative modelling of molecules and materials"
_ICML.cc/2025/Conference — ICML 2025 poster_

### Official Review · Reviewer_7DhL · 2025-02-26

**Overall Recommendation:** 3

**Summary:**

This paper proposes a new model based on all-atom Diffusion with transformers, for generating both periodic crystals and non-periodic molecules. Experiments are performed on standard benchmarks for these appications.  The authors show how this model helps speedup standard equivariant diffusion models and how it can benefit from scaling.

**Claims And Evidence:**

the claims are well supported.

**Essential References Not Discussed:**

given that the authors emphasize on the benefits of not having equivariance in their model, the authors could cite other work that also removed equivariance constraints in generative models for molecules / materials e.g. "Language models can generate molecules, materials, and protein binding sites directly in three dimensions as xyz, cif, and pdb files" Flam-Shepherd et al. 2023 or other work for unconditional molecule generation e.g. "3D molecule generation by denoising voxel grids" Pinheiro et al 2023. This section of related work without equivariance is missing in the related work, yet the authors did mention AlphaFold3 and Wang et al 2024 which are also relevant work on this front, not explicitly on generation.

**Experimental Designs Or Analyses:**

yes

**Methods And Evaluation Criteria:**

using a transformer for building a unified model for molecule and materials makes a lot of sense; moreover, replacing equivariance-based constraints by data augmentation to leverage more expressive architecture is also a recent trend in this field.

**Other Comments Or Suggestions:**

see above

**Other Strengths And Weaknesses:**

overall, I found the paper well executed with a nice idea of unifying different generative tasks; in particular the choices on the ML front, follow many recent trends in generative models which make a lot of sense. yet the paper is only focused on extremely small systems (up to 10 atoms as mentioned in the conclusion); arguably this is a simple setting. I would recommend the authors to include some experiments on some more challenging tasks; in particular staying with small molecule generation, GEOM-drugs is a more challenging setting for which comparison to other baselines would be more convincing. Moreover, I recommend the authors to extend the metrics in the small molecule front e.g. following the framework of MiDi: Mixed Graph and 3D Denoising Diffusion for Molecule Generation for which several models are compared agains.

**Questions For Authors:**

please address the limitations on the scale of the datasets and metrics of small molecule generation.

**Relation To Broader Scientific Literature:**

the paper proposes another approach to generate both material and molecules; as such the idea of learning a model across modalities is interesting and fairly new.

**Theoretical Claims:**

there were no theoretical claims

---

> ### Author Rebuttal · Authors · 2025-03-27
>
> > small systems - simple setting…include experiments on more challenging tasks
>
> ---
>
> **[In our response to Reviewer WqZ3](https://openreview.net/forum?id=89QPmZjIhv&noteId=RrA8d9t9eq), we presented results for GEOM based on your suggestion. ADiT continues to show strong results, generating physically realistic molecules and outperforming baselines.**
>
> ---
>
> We agree that systems from QM9 and MP20 are small, but they are complex and challenging. Doing well on them requires models to learn something about the underlying physics of atomic interactions.
>
> Let’s take MP20 as an example:
> - MP20 is restricted to up to 20 atoms in a unit cell because this is practically most relevant and largely covers the Materials Project distribution of known materials. Each atom's local environment is complex due to periodicity.
> - The S.U.N. rate metric for crystals/MP20 is a very challenging and practically relevant metric for materials discovery. ADiT has **improved S.U.N. rate from 4-5% of previous works up to 6.5%** (see Table 4). ADiT significantly outperformed the recently published MatterGen (Nature, 2024). Previous works showing smaller improvements have been published at top conferences (most recent example: FlowLLM at NeurIPS, 2024). There is a large community of researchers who care about S.U.N. rate on MP20.
>
> Ultimately, the goal of this paper was to introduce the **first unified architecture** for generative modelling of 3D atomic systems, and to demonstrate **transfer learning** across periodic crystals and non-periodic molecular systems. We believe this will make a novel and valuable contribution to the ICML community.
>
> Also note that our best models are based on pure Transformers - which have shown to be extremely scalable to large-scale inputs - so there is nothing inherent to our methodology that would prevent scaling up to larger systems or more samples.
>
> > extremely small systems - up to 10 atoms
>
> Just to clarify the details here:
> - MP20: **20 atoms** in a periodically repeating unit cell. This means that each atom can **interact with a far greater number of atoms** than the size of the unit cell due to periodicity.
> - QM9: 9 heavy atoms, which doesn’t include hydrogen. When including hydrogen (which is how our models are trained), systems go up to **20-30 atoms on average**.
>
> Which is why we wrote “tens of atoms”, not 10 atoms.
>
> > extend metrics for small molecules, following MiDi
>
> Our evaluation followed **[Symphony](https://arxiv.org/abs/2311.16199)**, the most recently accepted small molecule generation paper at ICLR 2024. We believe that **Symphony's evaluation metrics are an improvement upon MiDi**.
>
> Here are our justifications:
> - We have included the validity and uniqueness metrics from MiDi/EDM. We found ADiT to outperform EDM and GeoLDM, both of which do better than MiDi.
> - Instead of atom and molecule stability metrics from MiDi/EDM which based on simple valency heuristics, Symphony used the **PoseBusters** metrics suite which provides information about the physical realism of generated molecules. PoseBusters provides a more holistic set of metrics using force field relaxations and physics-based checks instead of relying on heuristics.
> - PoseBusters also includes metrics for bond distances, angles, rings, and geometries, which supersede MiDi's histogram metrics and are more interpretable (in our opinion).
> - PoseBusters is now widely known and used in academia and industry, e.g. AlphaFold3 used PoseBusters.
>
> > "the paper proposes another approach to generate both material and molecules; as such the idea of learning a model across modalities is interesting and fairly new"
>
> The goal of this paper was to introduce the **first unified architecture** for generative modelling of 3D atomic systems, and to demonstrate **transfer learning** across modalities.
>
> We disagree with your characterisation that our work is just “another” approach as we are not aware of any other generative architecture that jointly generates periodic and non-periodic systems. We believe this will make a novel and valuable contribution to the ICML community, and would kindly request you to reconsider your position on our novelty.
>
> > citing Flam-Shepherd et al. 2023 and Pinheiro et al 2023
>
> We will definitely cite and discuss both papers in the Related Work - thanks for the pointers. We will also cite and discuss the MiDi paper and metrics in our revision.
>
> (Note that this year, ICML does not allow us to upload an updated PDF.)

---

> > ### Comment · Reviewer_7DhL · 2025-04-02
> >
> > thank you for the rebuttal and for the additional experiment on GEOM-drugs; for this dataset, the results would be more convincing if they included more recent baselines; EDM is far from the state-of-the-art on this dataset and most approaches now significantly outperform it. including the posebusters metrics for MiDi would be more convincing than a comparison to EDM. nevertheless, I will increase my score but this point still remains

---

> > > ### Author Response · Authors · 2025-04-03
> > >
> > > > posebusters metrics for MiDi
> > >
> > > Unfortunately, the MiDi author Clement Vignac's official checkpoints were deleted from his university's Google Drive when he moved to industry. The README for their codebase states this: "Update (July 2024): My drive account has unfortunately been deleted, and I have lost access some checkpoints. If you happen to have a downloaded checkpoint stored locally, I would be glad if you could send me an email at vignac.clement@gmail.com or raise a Github issue."
> > >
> > > As a result, checkpoints trained independently by another researcher, Ian Dunn, are now being shared by Clement on the MiDi Gtihub repo. However, we obtained worse results than the numbers reported in Clement's paper when running evaluation (using MiDi codebase as well as our codebase) on samples generated by Ian's checkpoint. A similar claim was made in [this paper](https://arxiv.org/pdf/2309.17296#page=18.44) (Appendix A.6): "We could not reproduce the results reported in the paper. We also re-evaluated the checkpoint given on GitHub and again could not confirm the reported results."
> > >
> > > This is why we chose to report the (subset of) metrics directly from the MiDi paper, and not the ones we re-computed.
> > >
> > > > include more recent baseline
> > >
> > > We used MiDi as you specifically mentioned MiDi in your review.
> > >
> > > We found two other methods which claim to outperform MiDi, but could not find their weights to re-evaluate their claims:
> > > - EquiGAT-Diff - "Navigating the Design Space of Equivariant Diffusion-Based Generative Models for De Novo 3D Molecule Generation" - https://arxiv.org/pdf/2309.17296
> > >     - They have provided inference code, but they only provide model checkpoints upon request. They state: "Currently, we provide trained model weights only upon request. Please reach out to ... if you are interested."
> > >     - We have contacted them but have yet to receive the weights.
> > > - "SemlaFlow – Efficient 3D Molecular Generation with Latent Attention and Equivariant Flow Matching" - https://arxiv.org/pdf/2406.07266
> > >     - There's no github link or codebase included in this paper.
> > >
> > > EquiGAT-Diff obtained a reported GEOM validity rate 94.6%, and SemlaFlow obtained 93.9%. Our ADiT-S model at validity rate 93.4% is able to reach roughly the same level of performance as MiDi and related equivariant models. (Note that ADiT-S is our smallest model and the result was obtained on one GPU during the brief rebuttal period time, without any hyperparameter optimization at all.)
> > >
> > > We think that these results demonstrate that the ADiT architecture can scale to larger datasets and larger system sizes, which is what your original concern was regarding.

---

### Official Review · Reviewer_KRy3 · 2025-03-11

**Overall Recommendation:** 3

**Summary:**

The authors introduce All-atom Diffusion Transformer (ADiT), a framework aimed at unifying latent diffusion approach across different spatial molecular structure modalities. The proposed method focuses specifically on small molecules and crystals. ADiT leverages a combination of joint variational autoencoder and latent diffusion transformer models to generate these molecular structures. The latent diffusion model constructs atom-wise latent codes, which the decoder then maps into atom descriptions, positions, and, optionally, crystal lattice parameters. The authors evaluate their approach on unconditional small molecule and crystal generation to benchmark its effectiveness.

**Claims And Evidence:**

The overall claims in the paper appear valid and reasonable. However, I recommend avoiding unjustified claims of primacy, particularly in lines 103-105: “our work is the first to develop unified generative models for both periodic and non-periodic atomic systems”. Preliminary approach [a] has proposed a unified language model for small molecule and crystal generation, as well as protein pocket generation.

a. Language models can generate molecules, materials, and protein binding sites directly in three dimensions as XYZ, CIF, and PDB files, Flam-Shepherd et al., 2023

**Essential References Not Discussed:**

As mentioned in the Claims and Evidence section, preliminary work [a] on multidomain spatial structure generation was not cited or compared against in the paper.

a. Language models can generate molecules, materials, and protein binding sites directly in three dimensions as XYZ, CIF, and PDB files, Flam-Shepherd et al.,  2023

**Experimental Designs Or Analyses:**

As discussed in “Methods and Evaluation Criteria”, the unconditional generation setups for molecules, crystals, and MOFs are reasonable and valid. The results from single-dataset and jointly trained models demonstrate the necessity of multidomain training. Additionally, the Ablation Study in the Appendix provides a justification for the chosen architectural design.

However, unconditional generation primarily serves as a proof-of-concept to demonstrate model capability, rather than its applicability to real-world tasks. A key question remains: Can the proposed approach be effectively adapted for conditional downstream tasks? For example, conformation generation [a], pocket-conditional generation [b], text description-conditioned generation [c], and MOF property optimization [d].

While the authors acknowledge the downstream tasks as future work in the Discussion section, the absence of at least one conditional setup or case study significantly reduces the paper’s impact. Including an experiment on conditional generation would strengthen the evaluation and provide clearer evidence of the model’s broader applicability.

a. Torsional Diffusion for Molecular Conformer Generation, Jing et al., 2022

b. 3D Equivariant Diffusion for Target-Aware Molecule Generation and Affinity Prediction, Guan et al., 2023

c. Fine-Tuned Language Models Generate Stable Inorganic Materials as Text, Gruver et al., 2024

d. MOFDiff: Coarse-grained Diffusion for Metal-Organic Framework Design, Fu et al., 2023

**Methods And Evaluation Criteria:**

The chosen evaluation metrics and baselines are appropriate and reasonable, and incorporating DTF and PoseBusters-derived metrics introduces a novelty into the evaluation of spatial small molecule structures generation. However, the choice of datasets raises concerns.
The QM9 and MP20 datasets contain relatively small structures (up to 9 heavy atoms in QM9 and up to 20 heavy atoms in MP20). In contrast, state-of-the-art generative models [a, b] for spatial small molecules unconditional generation typically benchmark against GEOM-DRUGS [c], which includes structures with up to 91 heavy atoms. For crystal generation, prior work [d] has used more complex datasets, such as MPTS-52 (up to 52 heavy atoms) and [e] used custom subsets with up to 30 heavy atoms, in addition to MP20.
While using QM9 and MP20 facilitates direct comparison with a broader range of baselines, incorporating larger and more complex datasets would provide deeper insights into the model’s capabilities and generalizability. The integration of larger structures dataset would further enhance the value of the proposed approach and allow comparison with state-of-the-art approaches on harder tasks.

a. MolDiff: Addressing the atom-bond inconsistency problem in 3D molecule diffusion generation, Peng et al., 2023

b. BindGPT: A Scalable Framework for 3D Molecular Design via Language Modeling and Reinforcement Learning, Zholus et al., 2024

c. Geom, energy-annotated molecular conformations for property prediction and molecular generation, Axelrod & Gomez-Bombarelli, 2022

d. FlowMM: Generating Materials with Riemannian Flow Matching, Miller et al., 2024

e. Fine-Tuned Language Models Generate Stable Inorganic Materials as Text, Gruver et al., 2024

**Other Comments Or Suggestions:**

1. I would recommend making Figure 2 smaller or moving it to the Appendix. Instead, it would be more useful to include examples of generated structures in the main text, as visual inspection of results would be beneficial for readers.

2. In Table 2, Validity results, I assume that one of the QM9-only ADiT entries should be marked with an asterisk *.

**Other Strengths And Weaknesses:**

The paper is well-written and easy to follow. The metrics effectively cover different aspects of spatial structure generation. The proposed modification to the latent diffusion framework is straightforward and focuses on small molecules and crystals, which, while limited, still represents a novel contribution. Additionally, the inference time efficiency of the approach, especially in comparison to pure diffusion models, is a strength of the method.

A minor weakness of the approach is the high number of lambda coefficients in the autoencoder loss. There is a lack of intuition on how to choose these coefficients and whether they are dataset-dependent.

**Questions For Authors:**

My concerns are addressed in the previous sections. The key issues are the lack of downstream tasks beyond unconditional generation and the use of relatively small datasets for small molecule and crystal generation.

**Relation To Broader Scientific Literature:**

The idea of applying the latent diffusion framework for generating chemical structures has been explored in prior work for 2D [a] and 3D molecules [b], as well as 3D proteins [c]. While the proposed modification to integrate crystal lattice parameters is relatively straightforward, I consider it to be a novel contribution of the paper.

a. 3M-Diffusion: Latent Multi-Modal Diffusion for Language-Guided Molecular Structure Generation, Zhu et al., 2024

b. Geometric Latent Diffusion Models for 3D Molecule Generation, Xu et al., 2023

c. A Latent Diffusion Model for Protein Structure Generation, Fu et al., 2023

**Theoretical Claims:**

N/A

---

> ### Author Rebuttal · Authors · 2025-03-27
>
> > relatively small structures - benchmark against GEOM
>
> ---
>
> **[In our response to Reviewer WqZ3](https://openreview.net/forum?id=89QPmZjIhv&noteId=RrA8d9t9eq), we presented results for GEOM based on your suggestion. ADiT continues to show strong results, generating physically realistic molecules and outperforming baselines.**
>
> ---
>
> We agree that systems from QM9 and MP20 are small, but they are complex and challenging. Doing well on them requires models to learn something about the underlying physics of atomic interactions.
>
> Let’s take MP20 as an example:
> - MP20 is restricted to up to 20 atoms in a unit cell because this is practically most relevant and largely covers the Materials Project distribution of known materials. Each atom's local environment is complex due to periodicity.
> - The S.U.N. rate metric for crystals/MP20 is a very challenging and practically relevant metric for materials discovery. ADiT has **improved S.U.N. rate from 4-5% of previous works up to 6.5%** (see Table 4). ADiT significantly outperformed the recently published MatterGen (Nature, 2024). Previous works showing smaller improvements have been published at top conferences (most recent example: FlowLLM at NeurIPS, 2024). There is a large community of researchers who care about S.U.N. rate on MP20.
>
> Ultimately, the goal of this paper was to introduce the **first unified architecture** for generative modelling of 3D atomic systems, and to demonstrate **transfer learning** across periodic crystals and non-periodic molecular systems. We believe this will make a novel and valuable contribution to the ICML community.
>
> Also note that our best models are based on **pure Transformers** - which have shown to be extremely scalable to large-scale inputs - so there is nothing inherent to our methodology that would prevent scaling up to larger systems or more samples.
>
> > MPTS-52
>
> Not applicable as MPTS is meant for evaluating **structure prediction**, where TS = temporal split, which is different from our de novo generation task.
>
> > unjustified claims of primacy - Flam-Shephard
>
> To the best of our knowledge, ADiT is the first **unified, jointly trained** generative model for periodic and non-periodic systems to demonstrate **transfer learning**.
>
> Flam-Shepherd trains 3 independent models on 3 different datasets with a different tokenisation strategy for each. They don't discuss how/whether their method can be applied for one unified model. Also, our results on MP20 are better than FlowLLM, another language model which outperforms Flam-Shepherd.
>
> We will discuss their paper in Related Work, as well as all other references you shared.
>
> > latent diffusion for generating chemical structures has been explored prior
>
> Our novelty is about **how** we used latent diffusion: For unification of system types (periodic and non-periodic systems together), as well as unification of mulit-modal data (categorical, numerical, floating point).
>
> We think this is new - **nobody has used latent diffusion in this unified manner** - and will be of interest to the ICML community.
> - [a] - Not directly related to 3D structure
> - [b] - They technically do latent diffusion, but the latents are still multi-modal (four latent scalars & one latent vector) - thus GeoLDM still uses equivariant diffusion, which is slow, and GeoLDM is not applicable beyond small molecules. Additionally, there are **several discrepancies** between the implementation reported in the paper and released on GitHub (most concerning: https://github.com/MinkaiXu/GeoLDM/issues/6). ADiT’s latent diffusion formulation is unified and faster, as well as outperforms GeoLDM.
> - [c] - Highly specific to proteins as latent representations are created by aggregating along the sequence
>
> (We will cite [a] and [c])
>
> > high number of lambda coefficients, lack of intuition
>
> The coefficients are needed for each of the different data types that constitute a 3D atomic system. The coefficients are **not dataset dependent**. The intuition for choosing them is simple: **“balance the relative magnitudes of the various losses”**. This is stated in the paper and is standard practice for supervised learning tasks like the VAE reconstruction used here.
>
> > conditional downstream tasks
>
> The focus of this paper on introducing the architecture and demonstrating transfer learning. Efforts for extensions to conditional tasks such as protein-pocket conditioned molecule generation (ie. SBDD) as well as property conditioned crystal generation (similar to MatterGen) are underway as independent papers. From a methods standpoint, adding conditioning is straightforward in Diffusion Transformers via classifier-free guidance tokens.
>
> We don’t feel that property conditioned small molecule generation, as benchmarked in papers like EDM/GeoLDM, is very practically relevant. We feel that it is unlikely that generating molecules with specific HOMO-LUMO gaps or dipole moments are relevant for drug discovery.
>
> > Suggestions
>
> We will implement both.

---

> > ### Comment · Reviewer_KRy3 · 2025-04-05
> >
> > I’m grateful to the authors for their responses to my questions and, in particular, for conducting the experiments on GEOM. However, I remain convinced that, despite the promising direction, the current form of the approach has limited applicability. The authors have only demonstrated results in the unconditional setting, whereas in practice, it is often crucial to control the properties of generated molecules and materials. Including one or two case studies on conditional generation would significantly strengthen the quality of the paper. Therefore, I have decided to keep my score unchanged.

---

> > > ### Author Response · Authors · 2025-04-05
> > >
> > > We've worked hard to address as many reviewer concerns in the rebuttal period as we possibly could.
> > >
> > > As we mentioned, this is an architecture focussed paper. It introduces a new architecture which can jointly generate periodic and non-periodic chemical systems. This is new and not possible with previous techniques. And we think this will be well received by the ICML and broader machine learning community.
> > >
> > > We've not made any claims about conditional design and practical applicability. We acknowledged this as a limitation in the Discussions section. And we want to address this limitation in future work. Adding more conditional experiments would not change the main contributions and claims of this paper (which is about introducing a new architecture, not its practical application yet).
> > >
> > > You already stated that the paper is well written, our claims are well supported, the methods and evaluations are appropriate, experiments are convincing, and the approach is overall promising. To us, as authors, it sounds like you support this paper based on the claims it makes, and do want to accept it - even though you think adding conditional experiments will further strengthen the paper.
> > >
> > > Please would you consider changing your vote from borderline to an accept if that is the case?
> > >
> > > ---
> > >
> > > P.S. Here are notable examples of papers on new molecular or crystal generative modelling architectures that were published at top conferences **without** showing experiments on conditional tasks:
> > >
> > > - FlowLLM - NeurIPS 2024 - https://arxiv.org/abs/2410.23405
> > > - Symphony - ICLR 2024 - https://arxiv.org/abs/2311.16199
> > > - MiDi - ECML 2023 - https://arxiv.org/abs/2302.09048
> > > - DiffCSP - NeurIPS 2023 - https://arxiv.org/abs/2309.04475
> > >
> > > Each of these papers brings new architectural ideas to the table and catalyzes further research into both the architectures as well as conditioning them for downstream practical applications.

---

### Official Review · Reviewer_yi4C · 2025-03-14

**Overall Recommendation:** 3

**Summary:**

The paper introduced a unified framework called All-atom Diffusion Transformer (ADiT) for generating periodic (crystals) and non-periodic (molecules) atomic systems. ADiT employs a two-stage approach: A Variational Autoencoder that maps atomic systems into a shared latent space, and a Diffusion Transformer generates decoded latent samples into valid structures. Evaluations on QM9 and MP20 benchmarks showed the effectiveness of the proposed approach.

**Claims And Evidence:**

Yes.

**Essential References Not Discussed:**

No.

**Experimental Designs Or Analyses:**

Yes.

**Methods And Evaluation Criteria:**

Yes.

**Other Comments Or Suggestions:**

No.

**Other Strengths And Weaknesses:**

**Strengths**:

- Unified framework: I think the idea of using a joint generative model for molecules and crystals is novel, avoiding domain-specific methods.
- Empirical performance: The paper showed strong empirical results outperforming domain-specific methods for crystals and molecule generations.

**Weaknesses**:

- Dataset limitations: The method is trained on relatively small datasets (QM9 ~ 130K molecules; MP20 ~ 45K crystals). As already mentioned by the authors, scalability on larger datasets is unexplored.
- Theoretical gaps: The paper lacks a theoretical justification for why a shared latent space works. More discussion on the latent space properties would be helpful.

**Questions For Authors:**

- Can you explain in more detail why joint training on MOFs reduces validity?

- The comparison to the equivariant baseline focuses on speed, but are there trade-offs in terms of equivariance properties? For example, does this affect the model's ability to respect the physical symmetries of crystals/ molecules?

**Relation To Broader Scientific Literature:**

The authors built their approach on several works in the literature, like Variational Autoencoder, Diffusion Transformer, and classifier-free guidance technique.

**Theoretical Claims:**

No.

---

> ### Author Rebuttal · Authors · 2025-03-27
>
> > trained on relatively small datasets - scalability on larger datasets
>
> ---
>
> **[In our response to Reviewer WqZ3](https://openreview.net/forum?id=89QPmZjIhv&noteId=RrA8d9t9eq), we presented results for the larger GEOM dataset of small molecules with up to 180 atoms. ADiT continues to show strong results, generating physically realistic molecules and outperforming baselines.**
>
> ---
>
> We agree that systems from QM9 and MP20 are small, but they are complex and challenging. Doing well on them requires models to learn something about the underlying physics of atomic interactions, and cannot be solved by simple approaches.
>
> Let’s take MP20 as an example:
> - MP20 is restricted to up to 20 atoms in a unit cell because this is practically most relevant and largely covers the Materials Project distribution of known materials. Each atom's local environment is complex due to periodicity.
> - The S.U.N. rate metric for crystals/MP20 is a very challenging and practically relevant metric for materials discovery. ADiT has **improved S.U.N. rate from 4-5% of previous works up to 6.5%** (see Table 4). ADiT significantly outperformed the recently published MatterGen (Nature, 2024). Previous works showing smaller improvements have been published at top conferences (most recent example: FlowLLM at NeurIPS, 2024). There is a large community of researchers who care about S.U.N. rate on MP20.
>
> Ultimately, the goal of this paper was to introduce the **first unified architecture** for generative modelling of 3D atomic systems, and to demonstrate **transfer learning** across periodic crystals and non-periodic molecular systems. We believe this will make a novel and valuable contribution to the ICML community.
>
> Also note that our best models are based on **pure Transformers** - which have shown to be extremely scalable to large-scale inputs - so there is nothing inherent to our methodology that would prevent scaling up to larger systems or more samples.
>
> > lack of theoretical justification for why a shared latent space works
>
> There are strong theoretical justifications of our choices:
>
> - **The underlying physics of atomic interactions is the same across all 3D atomic systems.** Interatomic distance between a carbon atom double bonded to an oxygen atom will be the same, whether they are part of a molecule or a crystal structure. Thus, a shared/unified latent space enables the model to learn shared principles of interatomic interactions.
> - Using a shared latent space **unifies system modalities** (both periodic and non-periodic atomic systems embedded together), as well as **unifies multi-modality data** (categorical, numerical, floating point embedded together). This makes it very easy to train a simple Gaussian diffusion model on the latent representations, instead of more complex equivariant diffusion formulations.
> - The advantage of jointly embedding periodic & non-periodic systems for ML force fields, aka. *universal interatomic potentials* was shown by recent works like JMP and MACE. More broadly, all of AI research is moving towards **joint training of large models on multiple datasets**, ie. learning unified latent representations.
>
> > trade-offs of equivariance? ability to respect physical symmetries?
>
> When developing ADiT, we ablated the impact of enforcing equivariance on the model -- **see Appendix D**. We tried to be pragmatic about whether or not to use equivariance.
>
> The non-equivariant version **improved generative performance** and **physical realism** of the crystals/molecules, as judged by evaluation metrics explicitly focussed on physics-based tests s.a. PoseBusters suite for molecules, and DFT-based rate for crystals.
>
> Overall, we and others have found that equivariance is not strictly necessary for training a good generative model. Perhaps it can even be an advantage: if you start with two rotated versions of a noisy Gaussian and non-equivariant denoising leads to two different (but equally valid) molecules/crystals, that gives you a more diverse generative model without sacrificing on performance!
>
> > joint training on MOFs reduces validity?
>
> Simple answer: training did not converge during the time period that we were allocated compute resources on a large enough GPU cluster for this experiment.
>
> Note that joint training lead to very strong validity for both crystals (91%) and molecules (95%), while reducing validity for MOFs. When looking at training dynamics (right side plot in Table 3), the model first learns crystals, then molecules, and finally starts learning MOFs later in training. We believe training for longer is likely to close the gap and potentially improve beyond the MOF-only variant of ADiT.
>
> Ultimately, our goal with including preliminary MOF results was to encourage the community to work further on MOFs - hybrid organic-inorganic materials for carbon capture and sustainability. We will release **open source code** and **datasets** which make it easy for others to build upon our initial experiments.

---

### Official Review · Reviewer_WqZ3 · 2025-03-23

**Overall Recommendation:** 4

**Summary:**

The authors note that current generative models for atomic systems - such as molecules and crystals - are are fragmented and overly specialized to the specific type of system they model. They propose all atom diffusion transformers (ADiT), which uses a two-step latent diffusion framework in which, first, mixed categorical and numerical data describing an atomic system are embedded into a shared latent space by training a VAE, and next, a diffusion transformer model is trained to model the latent distribution and generate new samples. Experimental results show that the proposed method effectively transfers knowledge between various types of atomic systems, outperforms baselines in various metrics, and scales in performance with increasing model capacity.

#### Update after rebuttal period
My review score remains the same.

**Claims And Evidence:**

The claims are supported.

**Essential References Not Discussed:**

N/A

**Experimental Designs Or Analyses:**

The experiments appear well-designed, and the analysis sound.

**Methods And Evaluation Criteria:**

The methods are valid.

**Other Comments Or Suggestions:**

N/A

**Other Strengths And Weaknesses:**

### Strengths

- The paper is well-written, and explains just enough of the physics and chemistry background needed for someone who is unfamiliar.
- The implementation details are very thorough, and clearly describe the steps needed to reproduce the proposed method.
- The proposed method is a promising approach not just for atomic systems, but also for other scientific domains that may have related data from heterogenous sources with mixed categorical and numerical traits.
- Empirical results are extensive and back up the major claims made by the authors.

### Weaknesses

- The authors refer to this point in their discussion, but it bears repeating that transfer of performance between the two related atomic systems that the authors mainly test does not necessarily indicate that the model has learned the underlying physics or that it can generalize well to very large-scale data.

**Questions For Authors:**

N/A

**Relation To Broader Scientific Literature:**

Generative models for atomic systems have thus far been very specialized for the specific application areas they model. This paper represents a unification of atomic system models based on the idea that the underlying physics in these systems should hold constant and generalize between seemingly different domains.

**Theoretical Claims:**

N/A

---

> ### Author Rebuttal · Authors · 2025-03-27
>
> Thank you for your positive comments and excellent summary of the work! Thanks for appreciating that the latent diffusion idea can be further applicable to multi-modality data in other scientific domains, too.
>
> > does not necessarily indicate that the model has learned the underlying physics
>
> ADiT obtains very strong results on both MP20 and QM9, exceeding the current state-of-the-art. These systems are small but they are complex and challenging. **Doing well on them requires models to learn something about the underlying physics of atomic interactions,** and cannot be solved by simple co-occurence/statistical approaches.
>
> Let’s take MP20 as an example:
> - MP20 is restricted to up to 20 atoms in a unit cell because this is practically most relevant and largely covers the Materials Project distribution of known materials. Each atom's local environment is highly complex due to infinite periodic tiling of a unit cell.
> - The S.U.N. rate metric for crystals/MP20 is a very challenging and practically relevant metric for materials discovery. ADiT has **improved S.U.N. rate from 4-5% of previous works up to 6.5%** (see Table 4). ADiT significantly outperformed the recently published MatterGen (Nature, 2024). Previous works showing smaller improvements have been published at top conferences (most recent example: FlowLLM at NeurIPS, 2024). There is a large community of researchers who care about S.U.N. rate on MP20.
>
> > or that it can generalize well to very large-scale data
>
> We agree, but the goal of this paper was to introduce the **first unified architecture** for generative modelling of 3D atomic systems, and to demonstrate **transfer learning** across periodic crystals and non-periodic molecular systems. We believe this will make a novel and valuable contribution to the ICML community.
>
> Below, we presented results for the larger GEOM dataset of small molecules with up to 180 atoms. ADiT continues to show strong results, generating physically realistic molecules and outperforming baselines.
>
> Also note that our best models are based on **pure Transformers** - which have shown to be extremely scalable to large-scale inputs - so there is nothing inherent to our methodology that would prevent scaling up to larger systems or more samples.
>
> ---
>
> # **TO ALL REVIEWERS - PLEASE READ BELOW - RESULTS ON GEOM**
>
> ---
>
> To demonstrate the scalability of ADiT to larger systems, we have run experiments on GEOM, as suggested by Reviewers KRy3 and 7DhL. GEOM includes 430,000 unique molecules and consists of larger systems than QM9, up to 180 atoms. Our experiments and evaluation follows the EDM/MiDi paper, with additional PoseBusters physics-based tests. Model setup and hyperparameters used are exactly as described in the paper.
>
> | Metric  | EDM | MiDi | ADiT-S
> | --- | --- | --- | --- |
> | Validity | 87.8 | 77.8 | 93.4
> | Uniqueness | 99.9 | 100.0 | 100.0
> | Atoms connected | 41.4 | 90.0 | 94.9
> | Bond angles | 91.8 | - | 96.2
> | Bond lengths | 90.2 | - | 96.8
> | Ring flat | 99.0 | - | 100.0
> | Double bond flat | 98.2 | - | 99.9
> | Internal energy | 89.2 | - | 94.2
> | No steric clash | 85.2 | - | 93.5
>
> ADiT-S (32M) outperforms or matches EDM and MiDi across all metrics. ADiT generates physically realistic molecules based on all PoseBusters criteria. Notably, molecules generated by ADiT are significantly more likely than EDM/MiDi to be connected as measured by the 'Atoms connected' score (measures % of molecules where there exists a path along bonds between any two atoms).
>
> We have not trained the larger ADiT-B and ADiT-L models due to resource constraints in the short rebuttal period. We expect larger models to further improve performance based on the scaling trends we have seen on QM9 and MP20.

---

> > ### Comment · Reviewer_WqZ3 · 2025-04-04
> >
> > Many thanks to the authors for addressing my main concern regarding the generalization of the proposed model. After reading your explanation and some of the related works you linked in your rebuttal, I now agree that your model appears to learn the underlying physics across heterogenous atomic systems.

---

> > > ### Author Response · Authors · 2025-04-05
> > >
> > > Thank you for championing this paper.

---

### Decision · Program_Chairs · 2025-05-01

**Decision:**

Accept (poster)

**Comment:**

This is an interesting paper addressing an important area of research. The reviewers are generally supportive to this paper, esp. after author rebuttals. Thus an accept is recommended.